# Ice-nucleating ability of aerosol particles and possible sources at three coastal marine sites

Meng Si[1], Victoria E. Irish[1], Ryan H. Mason[1], Jesús Vergara-Temprado[2], Sarah Hanna[1], Luis A. Ladino[3*], Jacqueline D. Yakobi-Hancock[3], Corinne L. Schiller[4], Jeremy J. B. Wentzell[5], Jonathan P. D. Abbatt[3], Ken S. Carslaw[2], Benjamin J. Murray[2], Allan K. Bertram[1]

[1]Department of Chemistry, University of British Columbia, Vancouver, V6T1Z1, Canada
[2]Institute for Climate and Atmospheric Science, School of Earth and Environment, University of Leeds, Leeds, LS2 9JT, UK
[3]Department of Chemistry, University of Toronto, Toronto, M5S3H6, Canada
[*]Now at: Centro de Ciencias de la Atmósfera, Universidad Nacional Autónoma de México, Ciudad Universitaria, Mexico City, Mexico
[4]Air Quality Science Unit, Environment and Climate Change Canada, Vancouver, V6C3S5, Canada
[5]Air Quality Research Division, Environment and Climate Change Canada, Toronto, M3H5T4, Canada

*Correspondence to:* Allan Bertram (bertram@chem.ubc.ca)

**Abstract.** Despite the importance of ice-nucleating particles (INPs) for climate and precipitation, our understanding of these particles is far from complete. Here, we investigated INPs at three coastal marine sites in Canada, two at mid-latitude (Amphitrite Point and Labrador Sea), and one in the Arctic (Lancaster Sound). For Amphitrite Point, 23 sets of samples were analyzed, and for Labrador Sea and Lancaster Sound, one set of samples was analyzed for each location. At all three sites, the ice-nucleating ability on a per number basis (expressed as the fraction of aerosol particles acting as an INP) was strongly dependent on the particle size. For example, at diameters of around 0.2 µm, approximately 1 in $10^6$ particles acted as an INP at -25 ºC, while at diameters of around 8 µm, approximately 1 in 10 particles acted as an INP at -25 ºC. The ice-nucleating ability on a per surface area basis (expressed as the surface active site density, $n_s$) was also dependent on the particle size, with larger particles being more efficient at nucleating ice. The $n_s$ values of supermicron particles at Amphitrite Point and Labrador Sea were larger than previously measured $n_s$ values of sea spray aerosol, suggesting that sea spray aerosol was not a major contributor to the supermicron INP population at these two sites. Consistent with this observation, a global model of INP concentrations under-predicted the INP concentrations when assuming only marine organics as INPs. On the other hand, assuming only K-feldspar as INPs, the same model was able to reproduce the measurements at a freezing temperature of -25 ºC, but under-predicted INP concentrations at -15 ºC, suggesting that the model is missing a source of INPs active at a freezing temperature of -15 ºC.

## 1 Introduction

Aerosol particles are ubiquitous in the atmosphere, yet only a small fraction of these particles, referred to as ice nucleating particles (INPs), are able to initiate the formation of ice at temperatures warmer than homogeneous freezing temperatures.

INPs may impact the frequencies, lifetime, and optical properties of ice and mixed-phase clouds (Andreae and Rosenfeld, 2008; Cziczo and Abbatt, 2001; Lohmann and Feichter, 2005).

It is now well established that mineral dust particles represent a large fraction of INPs in the atmosphere (Hoose et al., 2010). For example, laboratory studies have shown that mineral dust particles are efficient at nucleating ice (Atkinson et al., 2013; Boose et al., 2016a; Broadley et al., 2012; Eastwood et al., 2008; Field et al., 2006; Hartmann et al., 2016; Hiranuma et al., 2015; Kanji and Abbatt, 2010; Knopf and Koop, 2006; Murray et al., 2011; Wex et al., 2014). Field measurements have shown that mineral dust is a main component of INPs at different locations (Boose et al., 2016b; DeMott et al., 2003; Klein et al., 2010; Prenni et al., 2009; Worringen et al., 2015). Modeling studies have also suggested that mineral dust particles are a major contributor to INP concentrations in many locations around the globe (Hoose et al., 2010; Vergara-Temprado et al., 2017).

Recent studies also suggest that sea spray aerosol may be an important source of INPs in some remote marine regions (Wilson et al., 2015). For example, field and laboratory measurements have shown that seawater contain particles that can nucleate ice (Alpert et al., 2011a, 2011b; Irish et al., 2017; Knopf et al., 2011; Schnell, 1977; Schnell and Vali, 1976, 1975; Wilson et al., 2015), and these INPs in seawater are thought to be emitted into the atmosphere by wave breaking and bubble bursting mechanisms (DeMott et al., 2016; Wang et al., 2015). Field measurements suggest that ambient INPs collected in marine environment can come from marine origin (DeMott et al., 2016; Rosinski et al., 1986, 1988; Schnell, 1982), and modeling studies have shown that sea spray aerosol is a major source of INPs in some remote marine environments (Burrows et al., 2013; Vergara-Temprado et al., 2017; Wilson et al., 2015). Modeling studies have also suggested that INPs from the ocean can significantly modify the properties of mixed-phase clouds in the atmosphere, with implications for radiative forcing predictions (Yun and Penner, 2013). Despite the growing evidence indicating that sea spray aerosol is an important type of INPs, our understanding of when and where sea spray aerosol is an important component of the total INP population is far from complete. Additional field measurements of INPs in marine environments would help improve our understanding of this topic.

Here we report INP measurements in the immersion mode from three coastal marine sites. Immersion freezing refers to freezing initiated by INPs immersed in liquid droplets (Vali et al., 2015), and this freezing mode is considered to be the most relevant for mixed-phase clouds (Ansmann et al., 2009; de Boer et al., 2011; Westbrook and Illingworth, 2011). The three coastal marine sites investigated were Amphitrite Point, Labrador Sea, and Lancaster Sound (Fig. 1). For two of these sites (Amphitrite Point and Labrador Sea), the size distributions of INPs in the immersion mode have been reported previously (Mason et al., 2015a, 2016). In the following, we build on these previous measurements by reporting the following for all three coastal marine sites: 1) the size distribution of INPs, 2) the fraction of aerosol particles acting as an INP as a function of size, and 3) the surface active site density, $n_s$, as a function of size. In addition, we compare the INP measurements to predictions from a recently developed global model of INP concentrations (Vergara-Temprado et al., 2017). We use this combined information to help determine if sea spray aerosol or mineral dust are the major sources of INPs at these three sites. This type of information is needed to help constrain future modeling studies of INPs and mixed-phase clouds.

## 2 Methods

### 2.1 Measurements of INP concentrations as a function of size

Concentrations of INPs as a function of size were measured with the micro-orifice uniform deposit impactor-droplet freezing technique (MOUDI-DFT; Mason et al., 2015b). This technique involves collecting size-fractionated aerosol particles on hydrophobic glass slides with a micro-orifice uniform deposit impactor (MOUDI; Marple et al., 1991), and determining the freezing properties of collected aerosol particles with the droplet freezing technique (DFT). Details are given below.

### 2.1.1 Aerosol particle sampling with a MOUDI

A MOUDI (model 110R or 120R; MSP Corp., Shoreview, MN, USA) was used to collect size-fractionated aerosol particles. Aerosol particles were sampled at a flow rate of 30 L min$^{-1}$. The MOUDI has eleven stages, and each stage consists of a nozzle plate and an impaction plate. Aerosol particles were collected by inertial impaction onto hydrophobic glass slides (HR3-215; Hampton Research, USA) positioned on top of each impaction plate. Custom substrate holders were used to position the glass slides within the MOUDI. See Mason et al. (2015b) for details on the substrate holders. Stages 2 through 8 of the MOUDI were analyzed for this study (seven stages in total), corresponding to aerodynamic diameters of 5.6-10 μm, 3.2-5.6 μm, 1.8-3.2 μm, 1.0-1.8 μm, 0.56-1.0 μm, 0.32-0.56 μm and 0.18-0.32 μm, respectively, where the bounds are 50 % cut-off efficiencies (Marple et al., 1991).

Particle rebound from the substrate is an issue when sampling particles with an inertial impactor. Rebound occurs when the kinetic energy of the particles striking the impactor substrate exceeds the adhesion and dissipation energies at impact (Bateman et al., 2014). Rebound can alter the number concentration and size distribution of the INPs determined with the MOUDI-DFT. Previous work has shown that particle rebound can be reduced when relative humidity (RH) is above 70 % (Bateman et al., 2014; Chen et al., 2011; Fang et al., 1991). In addition, good agreement between INP concentrations measured by the MOUDI-DFT and INP concentrations measured by a continuous flow diffusion chamber (a technique that is not susceptible to rebound) has been observed in previous field campaigns when the RH of the sampled aerosol stream was as low as 40-45 % (DeMott et al., 2017; Mason et al., 2015b).

### 2.1.2 Droplet freezing experiments

The freezing properties of the collected aerosol particles were determined using the DFT (Iannone et al., 2011; Mason et al., 2015b; Wheeler et al., 2015). Briefly, the hydrophobic glass slides with the collected particles were placed in a temperature- and humidity-controlled flow cell coupled to an optical microscope (Axiolab; Zeiss, Oberkochen, Germany). The temperature was decreased to approximately 0 ℃, and the relative humidity was increased to above water saturation using a humidified flow of He (99.999 %, Praxair), resulting in the condensation and growth of water droplets on the collected particles. On average, approximately 40 droplets were analyzed in each experiment. The final droplet size was approximately 50-150 μm in diameter, and the spacing between droplets was roughly 100 μm, on average. After the formation of droplets,

the flow cell was cooled down to -40 ºC at a rate of -10 ºC min[-1] while images of the droplets were recorded. During this process, most freezing events occurred by immersion freezing, while approximately 10 % occurred by contact freezing, which refers to the freezing of liquid droplets caused by contact with neighbouring frozen droplets. When calculating INP concentrations, the contact freezing was accounted for in two ways: (i) an upper limit to the fraction frozen by immersion freezing was calculated by assuming all the contact freezing droplets froze by immersion freezing; (ii) a lower limit to the fraction frozen by immersion freezing was calculated by assuming all the contact freezing droplets remained liquid until the homogeneous freezing temperature was reached. The freezing temperature for each droplet was determined using the recorded images. From the freezing temperatures, the number of INPs active at a given temperature, #INPs(T), in each freezing experiment was calculated using the following equation:

$$\#INPs(T) = \left(-ln\left(\frac{N_u(T)}{N_0}\right)N_0\right)f_{nu,0.25-1\,mm}, \tag{1}$$

where $N_u(T)$ is the number of unfrozen droplets at temperature $T$; $N_0$ is the total number of droplets analyzed within an experiment; $f_{nu,0.25-0.1\,mm}$ is a correction factor for the non-uniformity of particle concentrations across the sample deposit at a scale of 0.25-0.1 mm (see Mason et al. (2015b) for details). Equation (1) accounts for the possibility of multiple INPs in one droplet (Vali, 1971).

The number concentration of INPs in the atmosphere, [INPs(T)], was then determined using the following equation:

$$[INPs(T)] = \#INPs(T)\left(\frac{A_{deposit}}{A_{DFT}V}\right)f_{nu,1\,mm}, \tag{2}$$

where $A_{deposit}$ is the total area of the sample deposit on each MOUDI impaction plate; $A_{DFT}$ is the area analyzed in the droplet freezing experiment; $V$ is the total volume of air sampled by the MOUDI; $f_{nu,1\,mm}$ is a correction factor for the non-uniformity of particle concentrations across the sample deposit at a scale of 1 mm (see Mason et al. (2015b) for details). The values of $f_{nu,1\,mm}$ and $f_{nu,0.25-0.1\,mm}$ are given in Table S1 in the Supplement.

## 2.2 Measurements of aerosol particle number and surface area size distributions

The combination of an aerodynamic particle sizer (APS) and a scanning mobility particle sizer (SMPS) was used to measure the aerosol number and surface area as a function of size. The APS (model 3321, TSI, Shoreview, MN, USA) measures diameters using the time-of-flight technique (Baron, 1986). At all three sites, the APS was operated with a sample flow of 1 L min[-1] and a sheath flow of 4 L min[-1]. The aerodynamic diameter range measured by the APS was 0.54-20 µm. Due to possible drop off in the sampling efficiency of the APS at sizes below 0.7 µm (Beddows et al., 2010), only APS data at sizes above 0.7 µm is used here, as done previously (Maguhn et al., 2003). The SMPS measures diameters based on the mobility of a particle in an electric field (Asbach et al., 2009; Hoppel, 1978). The SMPS was equipped with an inertial impactor at inlet that removed large particles outside the measurement range. At Amphitrite Point, the SMPS (model 3936, TSI) was operated at 0.57 L min[-1] sample flow with 2 L min[-1] sheath flow, and was used to measure particles with mobility diameters from 18.4 to 930.6 nm. At Labrador Sea and Lancaster Sound, the SMPS (model 3034, TSI) was operated at 1 L min[-1]

sample flow rate with 4 L min$^{-1}$ sheath flow and was used to measure particles with mobility diameters from 10 to 487 nm. The sampling condition and strategy is discussed below for each site.

## 2.3 Locations of sampling

Sampling occurred at three coastal marine sites: Amphitrite Point (48.92º N, 125.54º W) on Vancouver Island in British Columbia, Canada, Labrador Sea (54.59º N, 55.61º W) off the coast of Newfoundland and Labrador, Canada, and Lancaster Sound (74.26º N, 91.46º W) between Devon Island and Somerset Island in Nunavut, Canada (Fig. 1 and Table 1). All measurements were conducted as part of the NETwork on Climate and Aerosols: addressing key uncertainties in Remote Canadian Environments (NETCARE). The sampling dates, ambient RH values, ambient temperatures, and wind speeds during sampling are summarized in Table 1. Additional details about the three coastal marine sites are given below.

### 2.3.1 Amphitrite Point

Measurements at Amphitrite Point were carried out at a marine boundary layer site operated by Environment and Climate Change Canada, BC Ministry of the Environment, and Metro Vancouver. This site, which is frequently influenced by marine background air (McKendry et al., 2014), is located on the west coast of Vancouver Island, British Columbia, Canada, and is approximately 2.3 km south of the town of Ucluelet (population 1627), with the Pacific Ocean to its west and south, and Barkley Sound to its southeast and east.

MOUDI samples were collected from 6 to 27 August 2013 (18 day samples, 16 night samples) as part of a larger campaign that focused on cloud condensation nuclei and INPs at a marine coastal environment (Ladino et al., 2016; Mason et al., 2015b, 2015a; Yakobi-Hancock et al., 2014). The average INP concentrations as a function of size for the entire campaign have been reported previously as well as the INP concentrations for each sample (Mason et al., 2015a). In the following, we focus on a subset of these measurements (12 day samples, 11 night samples) corresponding to the time period when MOUDI-DFT, APS, and SMPS data are all available.

The MOUDI, APS, and SMPS were located within a mobile trailer (herein referred to as the NETCARE trailer) that was approximately 100 m from the rocky shoreline of the Pacific Ocean, separated by a narrow row of trees and shrubs approximately 2-10 m in height (Mason et al., 2015a). Aerosol particles were sampled through louvered total suspended particulate (TSP) inlets (Mesa Labs Inc., Butler, NJ, USA) that were approximately 25 m above sea level. The MOUDI and APS sampled directly from ambient air without drying, whereas the SMPS sampled ambient air through diffusion dryers. After MOUDI samples were collected, they were stored in petri dishes at room temperature and analyzed for INP concentrations within 24 h of collection.

Meteorological parameters were measured at a lighthouse that was approximately halfway between the NETCARE trailer and the Pacific Ocean. The ambient temperature and RH were measured with an HMP45C probe (Campbell Scientific, Logan, UT, USA). Wind speed was determined by a model 05305L Wind Monitor (R. M. Young, Traverse City, Michigan, USA). The temperature and RH within the NETCARE trailer were monitored using a temperature/RH sensor probe (Acurite

00891W3). The average temperature inside the NETCARE trailer during INP sampling period was 25 ºC, compared to an average ambient temperature of 14 ºC. As a result, the average RH of the air sampled by the MOUDI and APS inside the trailer was lower than ambient RH. Based on the average ambient temperature and RH and average temperature within the trailer, the average RH in the sampling line for the MOUDI and APS was approximately 50 %. Three successive diffusion

dryers were used prior to sampling with the SMPS, and the silica was exchanged and dried in an oven every 24 h. Although not measured on site in this campaign, this technique has been found to always reduce the RH to less than 20 %, and usually to less than 2 % (Ladino et al., 2014; Yakobi-Hancock et al., 2014). For typical atmospheric conditions, the equilibration timescale for gas-particle partitioning of semivolatile organic species is on the order of minutes to tens of minutes (Saleh et al., 2013). In contrast, the residence time in the dryers during sampling in the current study was approximately 10s.

Therefore, removal of semivolatile organic species during drying may not have been a large issue but cannot be completely ruled out.

### 2.3.2 Labrador Sea and Lancaster Sound

Measurements at Labrador Sea and Lancaster Sound were carried out onboard the Canadian Coast Guard Service (CCGS) vessel Amundsen. Amundsen serves as both an icebreaker for shipping lanes and a research vessel. The APS and MOUDI

were located next to each other on top of the bridge of this vessel. Sampling occurred through louvered TSP inlets that were approximately 15 m above sea level. The SMPS was positioned behind the bridge, approximately 20 m away from the APS and MOUDI, and sampled aerosol particles through 3/8" outside diameter stainless steel tube with an inverted U-shaped inlet that was approximately 15 m above sea level. Meteorological parameters were measured with sensors on a tower deployed on the foredeck of the Amundsen. Wind speed and direction were monitored at a height of 16 m above sea surface using a

conventional propeller anemometer (RM Young Co. model 15106MA). Temperature and RH were measured using an RH/Temperature probe (Vaisala model HMP45C212) housed in a vented sunshield.

One MOUDI sample was collected on 11 July 2014 while in the Labrador Sea off the coast of Newfoundland and Labrador. Results of this sample have been reported in Mason et al. (2016). A second MOUDI sample was collected on 20 July 2014 while in the Lancaster Sound between Devon Island and Somerset Island. During both sample collection periods, the

Amundsen was in transit, and the change of the coordinates was less than 0.5 degree in longitude, and less than 1 degree in latitude. When the two MOUDI samples were collected, the apparent wind direction was ±90º of the bow and the wind speed was > 9.3 km h$^{-1}$, suggesting that ship emissions did not influence the samples (Johnson et al., 2008). After collection, the samples were vacuum-sealed and stored in a 4 ºC fridge for 45-46 days prior to analysis. In contrast, the samples collected at Amphitrite Point were stored at room temperature and relative humidity for less than 24 h prior to INP analysis, as

mentioned above. Studies are needed to determine the effect of sample storage conditions on measured INP concentrations.

**2.4 Conversion of mobility diameter to aerodynamic diameter and corrections for hygroscopic growth**

At Labrador Sea and Lancaster Sound, a dryer was not used prior to sampling with the MOUDI, APS, and SMPS. Hence, for these two sites, all data correspond to the RH and temperatures during the sampling. The sizes measured by the MOUDI and the APS were in aerodynamic diameter, while the SMPS measured mobility diameter. To allow comparison between the INP data, APS data and SMPS data at these two sites, all the SMPS data has been converted to aerodynamic diameter (see Sect. S1 for details).

At Amphitrite Point, a dryer was also not used when sampling with the MOUDI and APS. On the other hand, dryers were used prior to sampling with the SMPS. To allow comparison between the INP data, APS data and SMPS data at this site, a free parameter was used to convert the SMPS data under dry conditions to aerodynamic diameters at the RH and temperature during the sampling. The free parameter was determined from the optimal overlap between the SMPS and APS data. This type of approach has been used successfully in the past to merge SMPS and APS data (Beddows et al., 2010; Khlystov et al., 2004) (see Sect. S2 for details).

**2.5 Back trajectory analysis**

For each MOUDI sample collected for INP analysis, a 3-day back trajectory was calculated using the HYSPLIT4 (Hybrid Single-Particle Lagrangian Integrated Trajectory) model of the NOAA Air Resources Laboratory (Stein et al., 2015). The GDAS (Global Data Assimilation System) 1 degree meteorological data was used as input. Back trajectories were initiated at the beginning of each MOUDI sampling period and at every hour until the end of the sampling period. The initiating height was the same as the height of the MOUDI sampling inlets as mentioned in Sect. 2.3. Back trajectories were also initiated at heights of 50 m and 150 m a.g.l. for each location to determine if the trajectories were sensitive to the height of initiation.

**2.6 Global model of INP concentrations**

A global model of INP concentrations relevant for mixed-phase clouds was used to predict concentrations of INPs at the three sampling sites (Vergara-Temprado et al., 2017). The model considers ice nucleation by K-feldspar, associated with desert dust, and marine organics, associated with sea spray aerosol, as INPs. In this model (GLOMAP-mode), aerosol number and mass concentration of several aerosol species are simulated in seven lognormal modes (3 insoluble and 4 soluble). The model has a horizontal resolution of 2.8 x 2.8 degrees with 31 vertical levels, and it is run for the year 2001 with meteorological fields from the European Centre for Medium-Range Weather Forecasts (ECMWF). Model output for the year 2001 was used since this model output was available from previous studies. The model includes a parameterization of boundary layer turbulence (Holtslag and Boville, 1993). The aerosol components are emitted internally mixed with the species of their mode, and several aerosol microphysical processes including new particle formation, particle growth, dry deposition and wet scavenging are represented (Mann et al., 2014). The INP concentrations are determined using a laboratory-based temperature-dependent density of active sites (active sites per unit surface area) for K-feldspar (Atkinson et

al., 2013) and a parameterization for marine organics based on the INP content of microlayer samples (expressed as active sites per unit mass of organic carbon) (Wilson et al., 2015) following the method shown in Vergara-Temprado et al. (2017) Appendix 2.

To predict INP concentrations at the three coastal marine sites, we used the output of the model for the grid cells that overlapped with the measurement locations. Since the measurements were carried out at the surface, output from the lowest level of the model was used. We calculated the mean concentrations of INPs from K-feldspar and marine organics for the months when measurements were made. For the simulations at Amphitrite Point, Labrador Sea, and Lancaster Sound, the months of August, July, and July were used, respectively.

As mentioned above and as done previously, the model output from the year 2001 was compared with measurements from different years. The inter-annual variability of aerosol concentrations simulated in the model is expected to be up to a factor of 2 due to differences in meteorological conditions (Marmer and Langmann, 2007). Model output for the year 2001 has been found to be able to reproduce the mass concentrations of mineral dust and marine organic aerosols within an order of magnitude with observations made in various years (Vergara-Temprado et al., 2017). Furthermore, the model output for the year 2001 was able to reproduce 62 % of the INP concentrations measured from studies spanning from 1973 to 2016 within an order of magnitude, which is the uncertainty in the predicted INP concentrations reported here (Fig. 8).

## 3 Results and discussion

### 3.1 Air mass sources from back trajectories

Figure 2 shows the 3-day back trajectories initiated for every hour during the MOUDI sampling at the three sites. Back trajectories initiated at heights of 50 m and 150 m a.g.l. (see Fig. S1-S2 in the Supplement) showed similar results. When considering all the back trajectories, at Amphitrite Point, 94 % of the time was spent over the ocean, at Labrador Sea, 40 % of the time was spent over the ocean, and at Lancaster Sound, 63 % of the time was spent over the ocean. The rest of the time was spent over the land. At Amphitrite Point, although the air masses were predominantly from the ocean based on the back trajectory analysis, the air masses did pass over local vegetation including coastal western hemlock. This local vegetation could potentially release enough INPs to overwhelm a small INP source from the ocean. Therefore, it was not possible to determine if the INPs are of marine or terrestrial origin based on the back trajectories alone. INPs may even have been long-range transported from sources that were not reached by the 3-day back trajectories (Vergara-Temprado et al., 2017).

### 3.2 INP concentrations as a function of size

In Fig. 3, the average INP number concentration is plotted as a function of size for the freezing temperatures of -15 ºC, -20 ºC, and -25 ºC. These three temperatures were chosen because freezing events were rare at temperatures warmer than -15 ºC, and for some MOUDI stages, all the droplets were frozen at temperatures lower than -25 ºC, making calculations of INP

concentrations using Eq. (1)-(2) not possible at temperatures lower than -25 ºC. Mason et al. (2015a) previously reported the average INP number concentrations as a function of size at Amphitrite Point for the time period of 6-27 August 2013. Here we report the average INP number concentrations as a function of size at the same site for a subset of the measurements (23 out of 34 samples) from Mason et al. (2015a) when both APS and SMPS data were available. Not surprisingly, the results

shown here are very similar to the results shown by Mason et al. (2015a). The result for Labrador Sea shown in Fig. 3 has also been reported previously in Mason et al. (2016), while the result for Lancaster Sound is new and represents the first report of INP concentrations as a function of size in the Arctic marine boundary layer. Lancaster Sound had the lowest INP concentrations among the three sites with average concentrations of INPs of 0 L$^{-1}$, 0.16 L$^{-1}$, and 0.67 L$^{-1}$ for the freezing temperatures of -15 ºC, -20 ºC, and -25 ºC, respectively. These numbers are consistent with several previous measurements

reported in the Arctic. For example, Mason et al. (2016) reported the following mean concentrations at a surface site in Alert, Nunavut: 0.05 L$^{-1}$, 0.22 L$^{-1}$ and 0.99 L$^{-1}$ for freezing temperatures of -15 ºC, -20 ºC, and -25 ºC, respectively. Bigg (1996) reported mean INP concentration of 0.01 L$^{-1}$ at -15 ºC on an icebreaker in the Arctic. Fountain and Ohtake (1985) measured mean INP concentrations of 0.17 L$^{-1}$ at -20 ºC at a surface site in Barrow, Alaska.

At Amphitrite Point and Labrador Sea, the majority of INPs measured were > 1 μm in diameter at all the temperatures

studied. At Lancaster Sound, the majority of INPs were also > 1 μm at -25 ºC. At -15 ºC, the concentrations of INPs were not above detection limit at any of the sizes, while at -20 ºC, freezing was only observed for sizes between 0.56 and 1 μm.

### 3.3 Size distributions of ambient aerosols

As mentioned above, the average concentrations of aerosol number and surface area as a function of size during sampling periods were determined from measurements with a SMPS and an APS. The results are shown in Fig. 4. The size

distributions were consistent with the size distributions measured at a mid-latitude North-Atlantic marine boundary layer site by O'Dowd et al., (2001) (see Fig. S3 in the Supplement). The average total number concentrations were 1487 ± 512 cm$^{-3}$, 3020 ± 128 cm$^{-3}$, and 946 ± 254 cm$^{-3}$ for Amphitrite Point, Labrador Sea, and Lancaster Sound, respectively. The number concentration at the Arctic site Lancaster Sound may have been influenced by the new particle formation in the summer Arctic marine boundary layer (Burkart et al., 2017; Tunved et al., 2013). For the size range of measured INPs (0.18-10 μm),

on average, < 3 % of the number concentration was supermicron in diameter, and < 47 % of the surface area concentration was supermicron in diameter.

### 3.4 Ice-nucleating ability on a per number basis

The ice-nucleating ability on a per number basis is represented as the fraction of aerosol particles acting as an INP. Shown in Fig. 5 is the fraction of aerosol particles acting as an INP as a function of size. To generate Fig. 5, first the aerosol number

concentrations (Fig. 4a) was binned using the same bin widths as the MOUDI, resulting in the total aerosol number concentration in each size bin (Fig. S4a). Then the INP concentration (Fig. 3) was divided by the aerosol number concentration (Fig. S4a). Figure 5 shows that the fraction of particles acting as an INP is strongly dependent on the size. For

Amphitrite Point and Labrador Sea, and for diameters of around 0.2 µm, approximately 1 in $10^6$ particles acted as an INP at -25 ºC. On the other hand, at the same sites and for diameters of around 8 µm, approximately 1 in 10 particles acted as an INP at -25 ºC. A similar trend may be present at Lancaster Sound, but at the smaller sizes investigated, the concentrations of INPs were below detection limit. The results in Fig. 5 show that the large particles at the three sites studied are extremely efficient at nucleating ice, and as a result, even though the number concentration of large particles might be small in the atmosphere, they can make an important contribution to the total INP number concentrations.

The strong dependence on the size shown in Fig. 5 is consistent with the small number of previous studies that investigated the fraction of aerosol particles acting as an INP as a function of size. Berezinski et al. (1988) studied INPs collected at 100-500 m above ground level in the southern part of the European territory of the former USSR. At a freezing temperature of -20 ºC and for a diameter of 0.1 µm, approximately 1 in $10^5$ particles acted as an INP, while for a diameter of 10 µm approximately 1 in 100 particles acted as an INP. A study of residuals of mixed-phase clouds by Mertes et al. (2007) found that 1 in 10 supermicron particles acted as an INP, while only 1 in $10^3$ submicron particles acted as an INP. Huffman et al. (2013) studied INPs collected at a semi-arid pine forest in Colorado, United States. At a freezing temperature of -15 ºC and for a diameter of 2 µm approximately 1 in $10^3$ particles acted as an INP, while at the same freezing temperature but for a diameter of 10 µm, more than 1 in 100 particles acted as an INP.

### 3.5 Surface active site density, $n_s$, as a function of size

The surface active site density, $n_s$, represents the number of ice nucleation sites per surface area (Connolly et al., 2009; Hoose and Möhler, 2012; Vali et al., 2015). This parameterization assumes that freezing is independent of time and can be scaled with surface area. Although these assumptions may not be accurate in all cases (Beydoun et al., 2016; Emersic et al., 2015; Hiranuma et al., 2015), $n_s$ is commonly used to describe freezing data due, in part, to its simplicity. Here we use the following equation to calculate $n_s$ as a function of size from our experimental data:

$$n_s(T) = \frac{[INPs(T)]}{S_{tot}},$$  (3)

where $[INPs(T)]$ is the INP concentration at temperature $T$ determined from Eq. (2) in a given size range, and $S_{tot}$ is the total surface area of all aerosol particles in the same size range. Since this equation considers the surface area of all aerosol particles, rather than the surface area of just the INPs, the calculated $n_s$ values correspond to the total atmospheric aerosol.

Shown in Fig. 6 are the measured $n_s$ values as a function of size determined with Eq. (3). To generate Fig. 6, first the aerosol surface area concentration (Fig. 4b) was binned using the same bin widths as the MOUDI, resulting in the total aerosol surface area concentration in each size bin (Fig. S4b). Following Eq. (3), the INP concentration (Fig. 3) was then divided by the total aerosol surface area concentration (Fig. S4b), resulting in $n_s$ values as a function of size. Figure 6 shows that $n_s$ is dependent on the size, with the larger particles being more efficient at nucleating ice. For Amphitrite Point and Labrador Sea, at a freezing temperature of -25 ºC, $n_s$ was approximately two orders of magnitude higher for 8 µm particles compared to 0.2 µm particles. The dependence of $n_s$ on size can be qualitatively explained by considering four different types of aerosol

particles each having progressively larger geometric mean diameters and larger $n_s$ values. As an example, consider a mixture of a) sulfate aerosols internally mixed with black carbon with a small $n_s$ and small geometric mean diameter, b) sea salt aerosols with larger $n_s$ and larger geometric mean diameter, c) clay particles with a larger $n_s$ and larger geometric mean diameter, and d) biological particles from terrestrial sources with the largest $n_s$ and largest geometric mean diameter. The assumption of a small $n_s$ for black carbon internally mixed with sulfate aerosols is consistent with previous measurements (e.g., Brooks et al., 2014; Chen et al., 2018).

To determine whether sea spray aerosol or mineral dust are the major sources of INPs at the three sites, the measured $n_s$ values were compared to $n_s$ values of sea spray aerosol and mineral dust at -15 ºC, -20 ºC, and -25 ºC, respectively (Fig. 7). The $n_s$ values of sea spray aerosol in Fig. 7 are from field studies in the marine boundary layer and laboratory studies of sea spray aerosol as reported in DeMott et al. (2016). Specifically, the data in Fig. 1A in DeMott et al. (2016) were re-plotted and fitted using linear regression (see Fig. S5 in the Supplement). Since the reported $n_s$ values in DeMott et al. (2016) were based on dry, geometric diameters, they overestimate the $n_s$ values based on wet, aerodynamic diameters at 95 % RH by a factor of 6 (see Sect. S3 in the Supplement). Figure 7 shows that the $n_s$ values of sea spray aerosol are smaller than the measured $n_s$ values in the supermicron range at all freezing temperatures at Amphitrite Point. This is also the case for Labrador Sea at freezing temperatures of -20 and -25 ºC. For Lancaster Sound, the $n_s$ values of sea spray aerosol are smaller than the measured $n_s$ values for sizes of 5.6-10 µm and a freezing temperature of -25 ºC. These combined results suggest that sea spray aerosol was not the major contributor to the supermicron INP population at Amphitrite Point and Labrador Sea, and not a major contributor to the largest INPs (5.6-10 µm in size) observed at Lancaster Sound.

The $n_s$ values of mineral dust particles shown in Fig. 7 are based on laboratory measurements with five different dust samples: Asian dust, Saharan dust, Canary Island dust, Israel dust, and Arizona test dust (Niemand et al., 2012). Specifically, the data in Fig. 6 in Niemand et al. (2012) were re-plotted and fitted using linear regression (see Fig. S6 in the Supplement). Since the reported $n_s$ values in Niemand et al. (2012) were based on geometric diameters, they overestimate the $n_s$ values based on aerodynamic diameters by a factor of 2 (see Sect. S3 in the Supplement). Figure 7 shows that the $n_s$ values for mineral dust are greater than or equal to the measured $n_s$ values at all three sites. These results suggest that mineral dust could be a possible source of the supermicron INPs at the three sites studied. However, these results do not confirm mineral dust as a major contributor of supermicron INPs nor do they rule out other types of particles as a major contributor of supermicron INPs. Note, the data from Niemand et al. (2012) corresponds to the $n_s$ values of only mineral dust particles, whereas the $n_s$ values reported here correspond to the total aerosol particles, as mentioned above. If we assume mineral dust particles are the only INPs in the atmosphere, and they account for 50 % of the total aerosol surface area, then the $n_s$ values of mineral dust shown in Fig. 7 divided by a factor of 2 would correspond to the $n_s$ values of the total atmospheric aerosol.

### 3.6 Comparison between measured and simulated INP concentrations

Shown in Fig. 8 is a comparison between the measured total INP concentrations (sum of the INP concentrations for all sizes measured) and the simulated INP concentrations at the surface at the three sites using a global model of INP concentrations

based on the ice nucleation of K-feldspar and marine organics. When considering only marine organics as INPs in the model, predicted INP concentrations are less than measured INP concentrations in all cases except for Amphitrite Point at a freezing temperature of -25 ℃. This suggests that sea spray aerosol is not the dominant source of INPs at the three coastal marine sites studied for all three temperatures, which is consistent with conclusions reached in Sect. 3.5. When considering only K-feldspar, associated with desert dust, as INPs in the model, the predicted INP concentrations at -25 ℃ are consistent with the measurements at all three sites, but at -15 ℃ and -20 ℃ the predicted INP concentrations are less than measured. When considering both marine organics and K-feldspar as INPs in the model the predicted INP concentrations at -25 ℃ are consistent with measurements, but at warmer temperatures, the predicted INPs are still less than measured. The underestimation of INP concentrations at warmer temperatures of the model could be explained by a missing source of INPs that are active at temperatures warmer than -25 ℃,  as hypothesized in Vergara-Temprado et al. (2017) based on the comparison with measurements at other sites. Possible sources missing in the model that could explain the high-temperature INPs include bacteria, fungal material, agricultural dust or biological nanoscale fragments attached to mineral dust particles (Fröhlich-Nowoisky et al., 2015; Garcia et al., 2012; Haga et al., 2013; Mason et al., 2015a; Möhler et al., 2008; Morris et al., 2004, 2013, O'Sullivan et al., 2014, 2015, 2016; Spracklen and Heald, 2014; Tobo et al., 2013, 2014).

Recently Mason et al. (2015a) investigated the source of INPs at Amphitrite Point using correlations between INP number concentrations, atmospheric particles, and meteorological conditions. Correlations between INP number concentrations and marine aerosol (sodium as a tracer) and marine biological activities (methanesulfonic acid as a tracer) were not statistically significant. On the other hand, a strong correlation was observed between INP concentrations and fluorescent bioparticles, suggesting biological particles from terrestrial sources were likely a dominant source of INPs at this site. These results are consistent and complementary to the studies presented above.

As discussed in Section 2.1.1, particle rebound from the substrate can be an issue when sampling particles with an inertia impactor. Good agreement between INP concentrations measured by the MOUDI-DFT and INP concentrations measured by a continuous flow diffusion chamber (a technique that is not susceptible to rebound) has been observed in previous field campaigns when the RH of the sampled aerosol stream was as low as 40-45 % (DeMott et al., 2017; Mason et al., 2015b). Nevertheless, particle rebound cannot be completely ruled out in the current study. If particle rebound was a factor when collecting particles with the MOUDI in the current study, the measured INP concentrations would be lower limits to the true INP concentrations, and the differences between simulated INP concentrations and measured INP concentrations shown in Fig. 8 would only be larger.

## 4 Summary and conclusions

The INP number concentrations in the immersion freezing mode as a function of size were determined at three coastal marine sites in Canada: Amphitrite Point (48.92º N, 125.54º W), Labrador Sea (54.59º N, 55.61º W), and Lancaster Sound (74.26º N, 91.46º W). For Amphitrite Point, 23 sets of samples were analyzed, and for Labrador Sea and Lancaster Sound,

one set of samples was analyzed for each location. The result for Lancaster Sound is the first report of INP number concentrations as a function of size in the Arctic marine boundary layer. The freezing ability of aerosol particles as a function of size was investigated by combining the size-resolved concentrations of INPs and the size distributions of aerosol number and surface area. We found that the fraction of aerosol particles acting as an INP is strongly dependent on the particle size. At -25 ºC and for Amphitrite Point and Labrador Sea, approximately 1 in $10^6$ particles acted as an INP at diameters around 0.2 µm, while approximately 1 in 10 particles acted as an INP at diameters around 8 µm. We also found that the surface active site density, $n_s$, is dependent on the particle size. At -25 ºC and for Amphitrite Point and Labrador Sea, $n_s$ was approximately two orders of magnitude higher for 8 µm particles compared to 0.2 µm particles. The size distribution of $n_s$ can be qualitatively explained by considering four different types of aerosol particles each having progressively larger geometric mean diameters and $n_s$ values.

Sea spray aerosol and mineral dust were investigated as the possible sources of INPs. Sea spray aerosol was not the major source of INPs based on the comparison of the measurements with the $n_s$ values of sea spray aerosol, and the INP concentrations predicted by a global model. On the other hand, the mineral dust may be a main source of INPs at the three sites and at a freezing temperature of -25 ºC based on the comparison of the measured INP concentrations with the predictions of a global model. However, the under-prediction of the INP concentrations at -15 ºC and -20 ºC suggests the existence of other possible sources of INPs such as biological particles from terrestrial sources or agricultural dust. Since only one sample was analyzed for both Labrador Sea and Lancaster Sound, additional samples should be collected and analyzed at these locations to determine the general applicability of the results presented here for these locations. In addition, since the results presented here correspond to surface measurements, similar studies as a function of altitude are needed to determine if these results are applicable to higher altitudes and to the free troposphere. Comparison with predictions of INPs from a high-resolution model would also be useful to assess the importance of local INP sources. Studies of the chemical composition of the INPs are also needed to test the conclusions reached in the current study.

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

**Table 1. The three sampling locations used in this study and conditions during sampling including ambient temperature (T) and relative humidity (RH). Included are the mean values and standard deviations. For Labrador Sea and Lancaster Sound, the coordinates are the locations at the mid-points of the sampling periods.**

| Location | Coordinates | Sampling dates | Sampling time (h) | Ambient T (ºC) | Ambient RH (%) | Wind speed (m/s) |
|---|---|---|---|---|---|---|
| Amphitrite Point, BC, Canada | 48.92º N, 125.54º W | 6-27 August 2013 | 7.8 ± 1.3 | 13.9 ± 1.0 | 97 ± 4 | 4.0 ± 2.3 |
| Labrador Sea, NL, Canada | 54.59º N, 55.61º W | 11 July 2014 | 6.2 | 10.9 ± 0.8 | 70 ± 5 | 5.4 ± 2.1 |
| Lancaster Sound, NU, Canada | 74.26º N, 91.46º W | 20 July 2014 | 5.3 | 2.8 ± 0.6 | 95 ± 1 | 4.6 ± 0.9 |

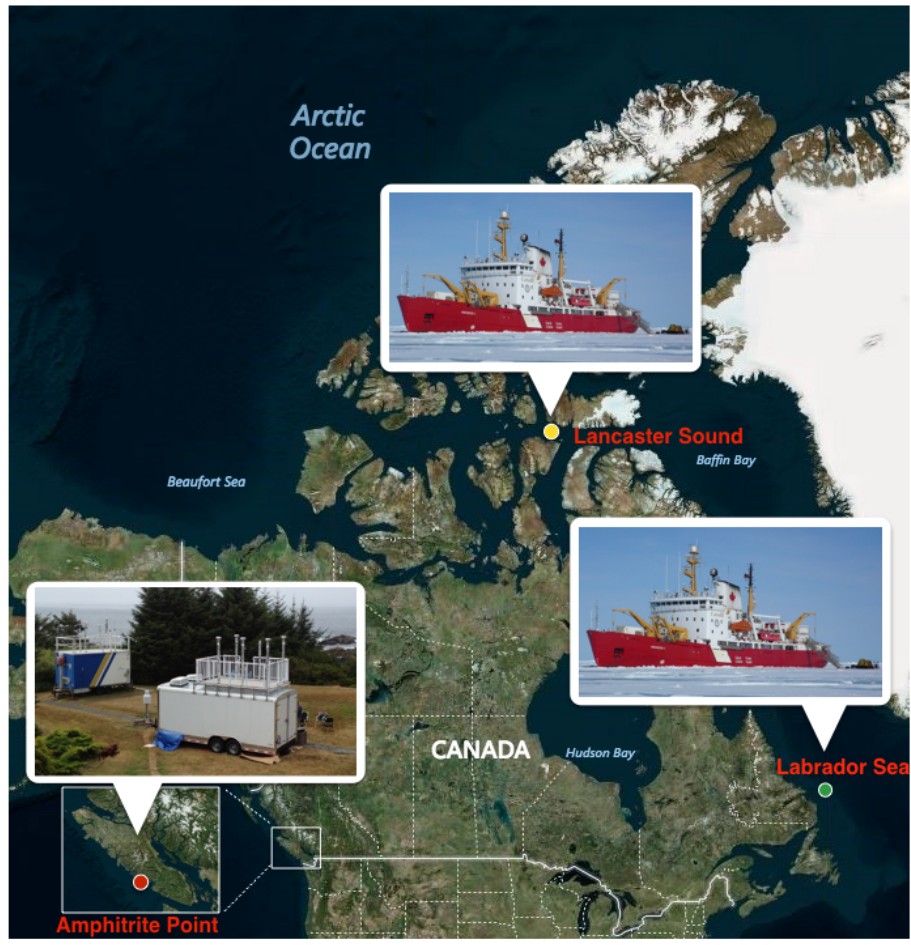

**Figure 1. Map showing the three sampling locations: Amphitrite Point (red dot), Labrador Sea (green dot), and Lancaster Sound (yellow dot). Inserts show the images of the sampling platform used at each location.**

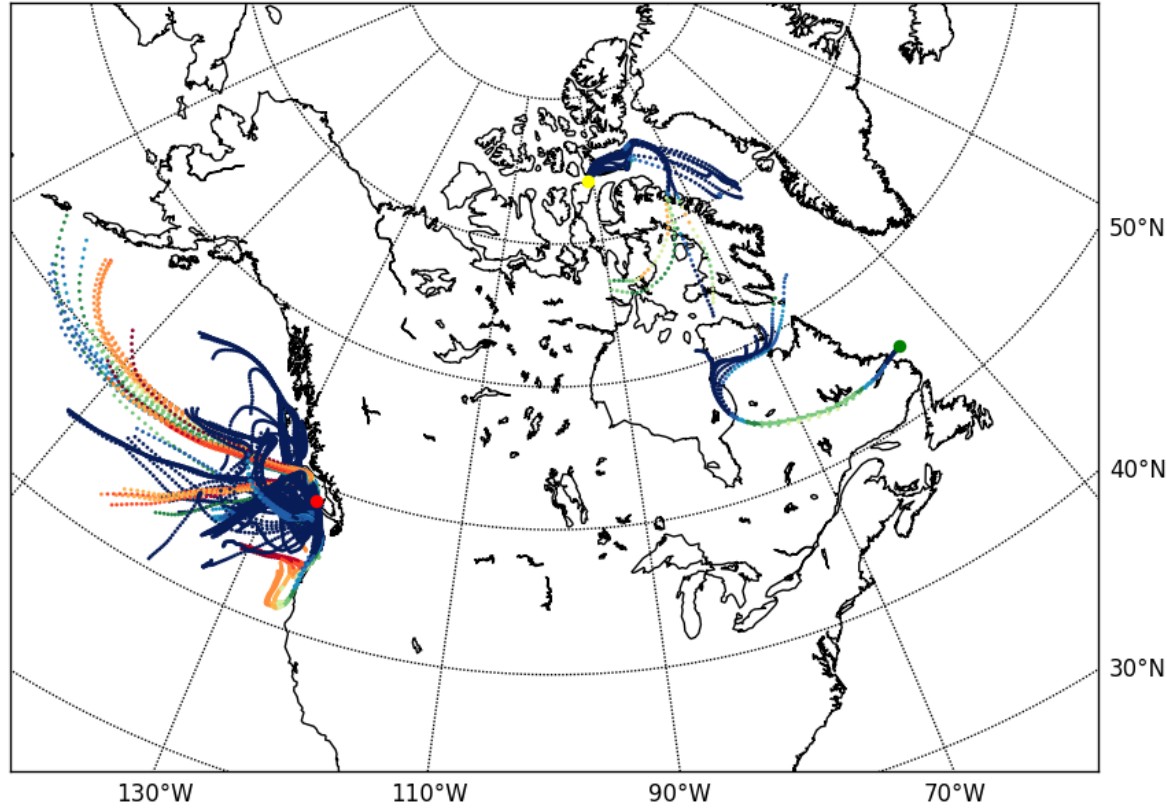

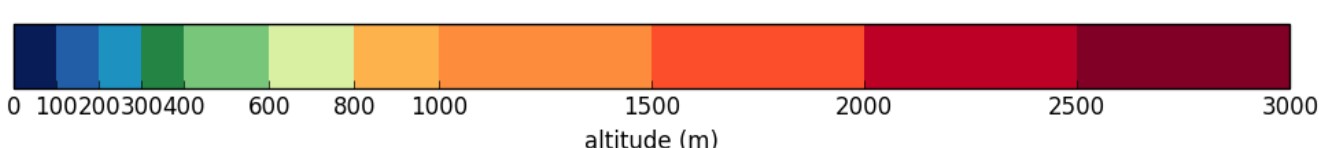

**Figure 2. The 3-day HYSPLIT back trajectories for Amphitrite Point (red dot), Labrador Sea (green dot) and Lancaster Sound (yellow dot). The back trajectories were calculated for every hour during the MOUDI sampling period. The altitude is indicated with the colour scale. Global Data Assimilation System (GDAS) meteorological data at 1° x 1° spatial resolution was used as input to calculate the back trajectories using HYSPLIT.**

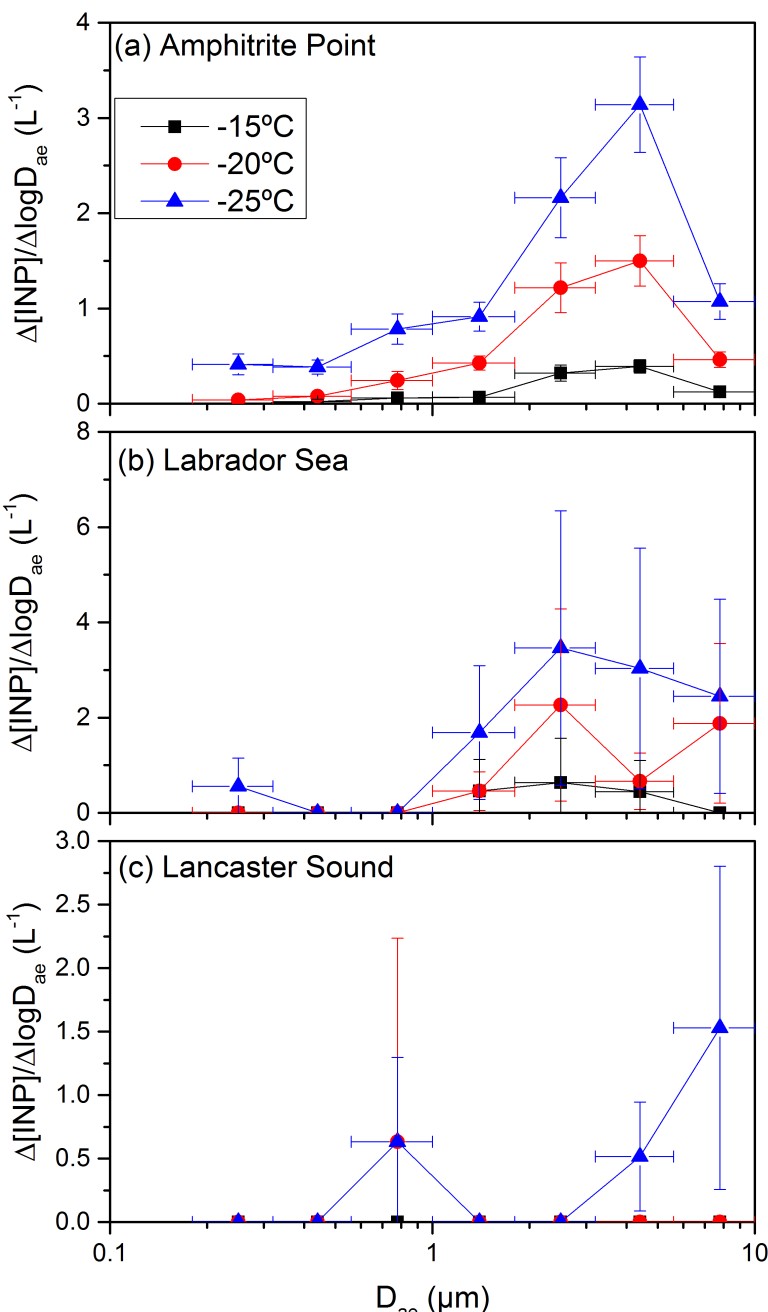

Figure 3. Average INP number concentrations at freezing temperatures of -15 °C, -20 °C, and -25 °C as a function of aerodynamic diameter ($D_{ae}$) for the three sites studied. The plotted x-values represent the midpoints of the size bins from the MOUDI. The x-error bars represent the widths of the size bins from the MOUDI. For the Amphitrite Point samples, standard error of the mean was used to represent the uncertainty of INP concentrations during the month. At both Labrador Sea and Lancaster Sound, only one MOUDI sample was collected, and we assume the monthly INP concentrations have the same normal distribution as the Amphitrite Point samples. Hence for the y-error bars at these locations, we assume the relative standard deviation for supermicron and submicron particles were the same as the relative standard deviation for supermicron and submicron particles observed in the Amphitrite Point data.

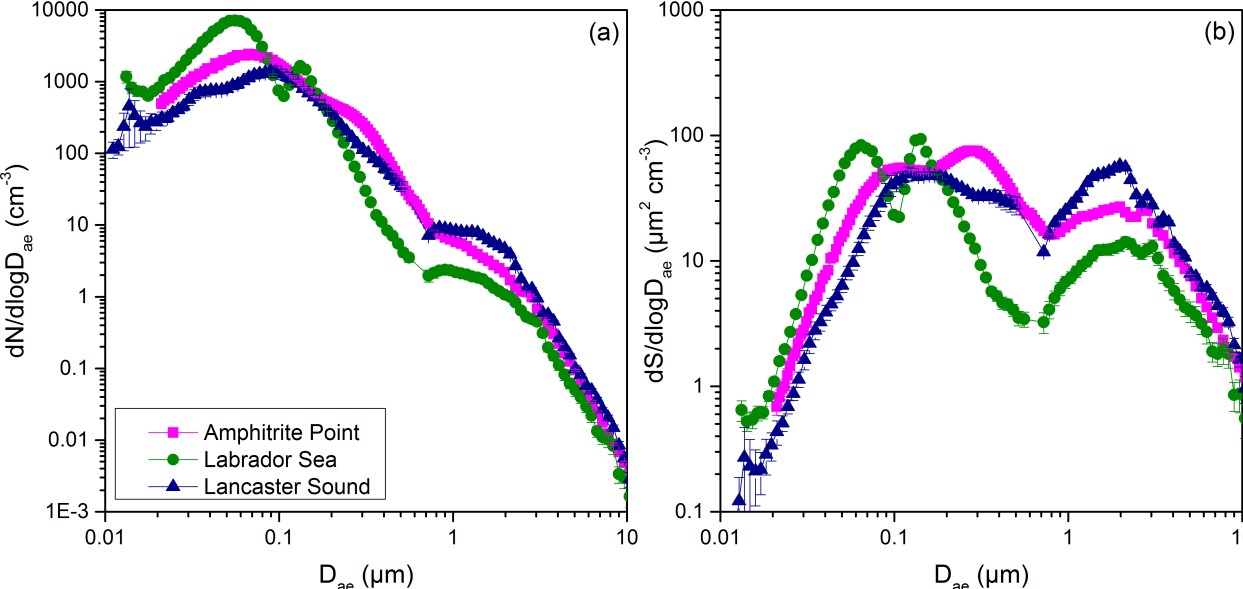

**Figure 4. Average concentrations of (a) aerosol number, *N*, and (b) surface area, *S*, as a function of aerodynamic diameter, *D$_{ae}$*. The y-error bars represent the standard error of the mean for each size bin. In many cases, the error bars are smaller than the size of the symbols. For cases where a gap existed between the SMPS data and the APS data, a straight line was used to extrapolate the data.**

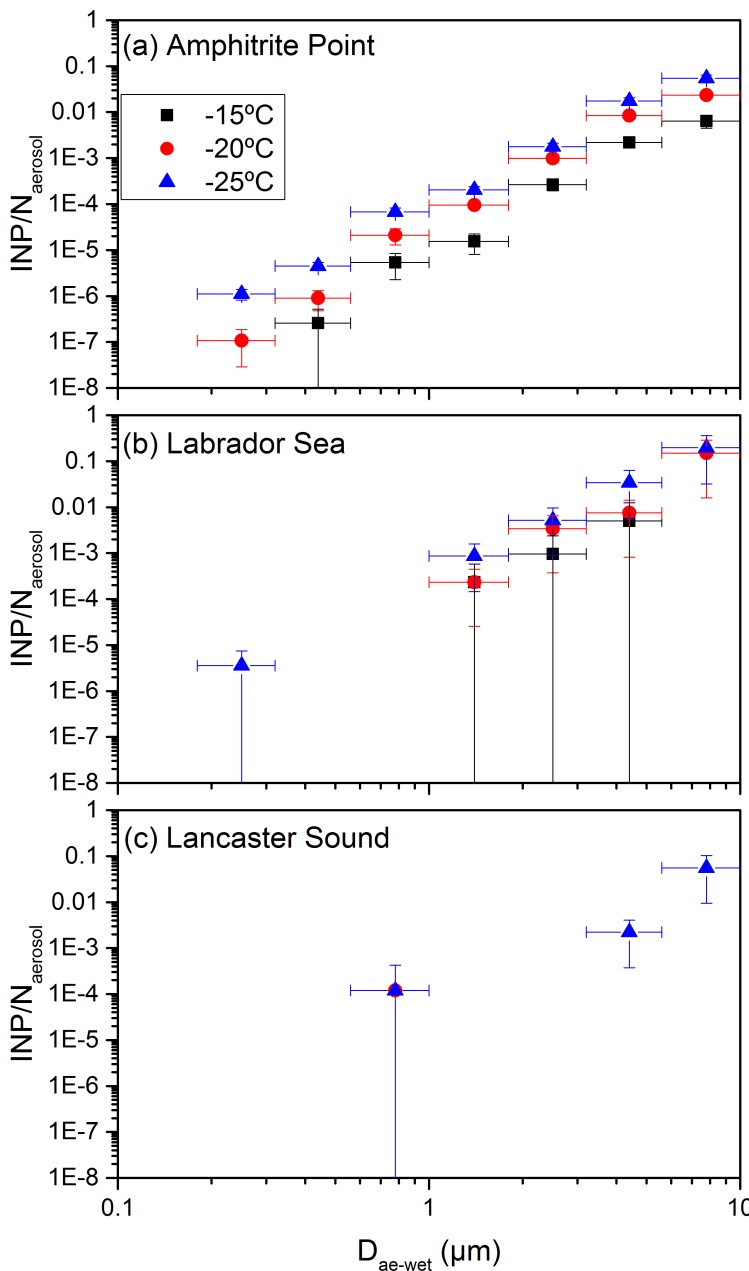

**Figure 5. The fraction of aerosol particles acting as an INP (*INP/N$_{aerosol}$*) plotted as a function of aerodynamic diameter (*D$_{ae}$*) at -15 ºC, -20 ºC, and -25 ºC, respectively, where *N$_{aerosol}$* is the number of aerosol particles in a given size bin The plotted x-values represent the midpoints of the size bins from the MOUDI. The x-error bars represent the widths of the size bins, and the y-error bars are the propagated uncertainties from INP concentrations as a function of size (Fig. 3) and aerosol number concentrations as a function of size (Fig. S4a). In some cases, the y-error bars are smaller than the size of the symbols.**

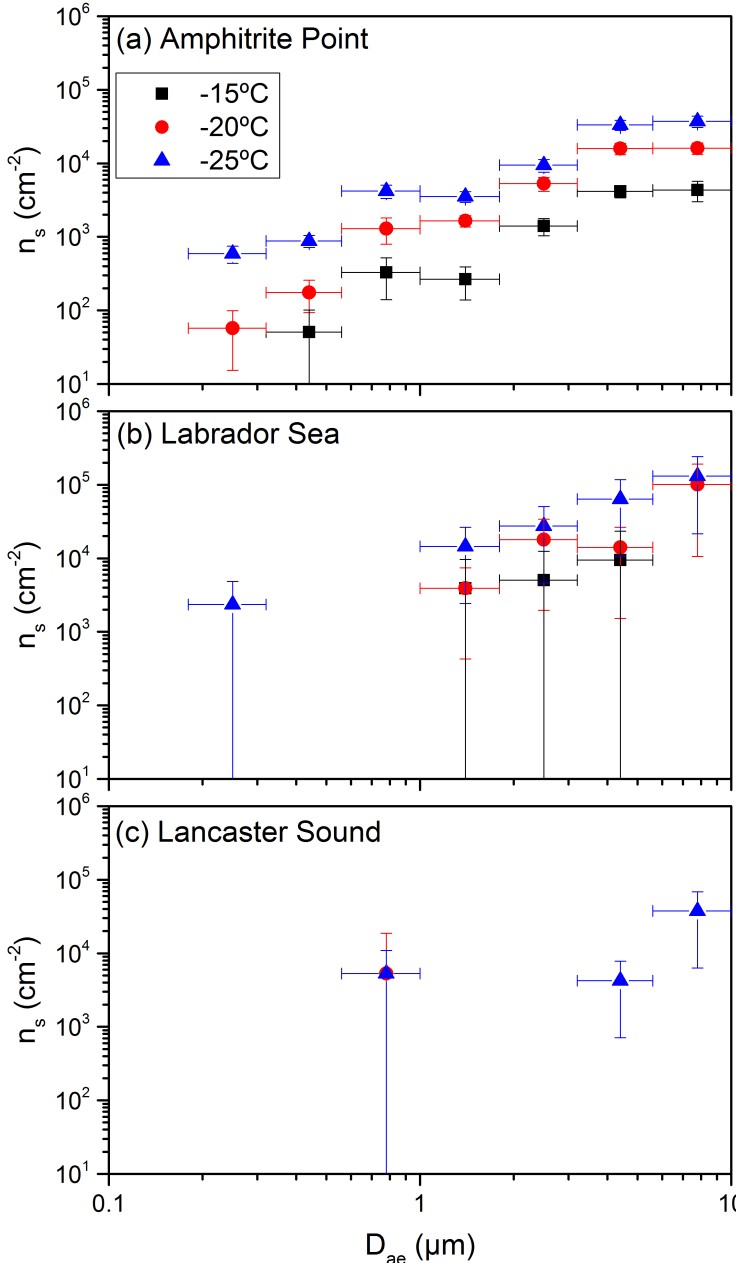

Figure 6. Surface active site density, $n_s$, as a function of aerodynamic diameter ($D_{ae}$) at -15 ºC, -20 ºC, and -25 ºC, respectively. The plotted x-values represent the midpoints of the size bins from the MOUDI. The x-errors represent the widths of the size bins, and the y-errors are the propagated uncertainties from INP concentrations as a function of size (Fig. 3) and aerosol surface area concentrations as a function of size (Fig. S4b). In some cases, the y-error bars are smaller than the size of the symbols.

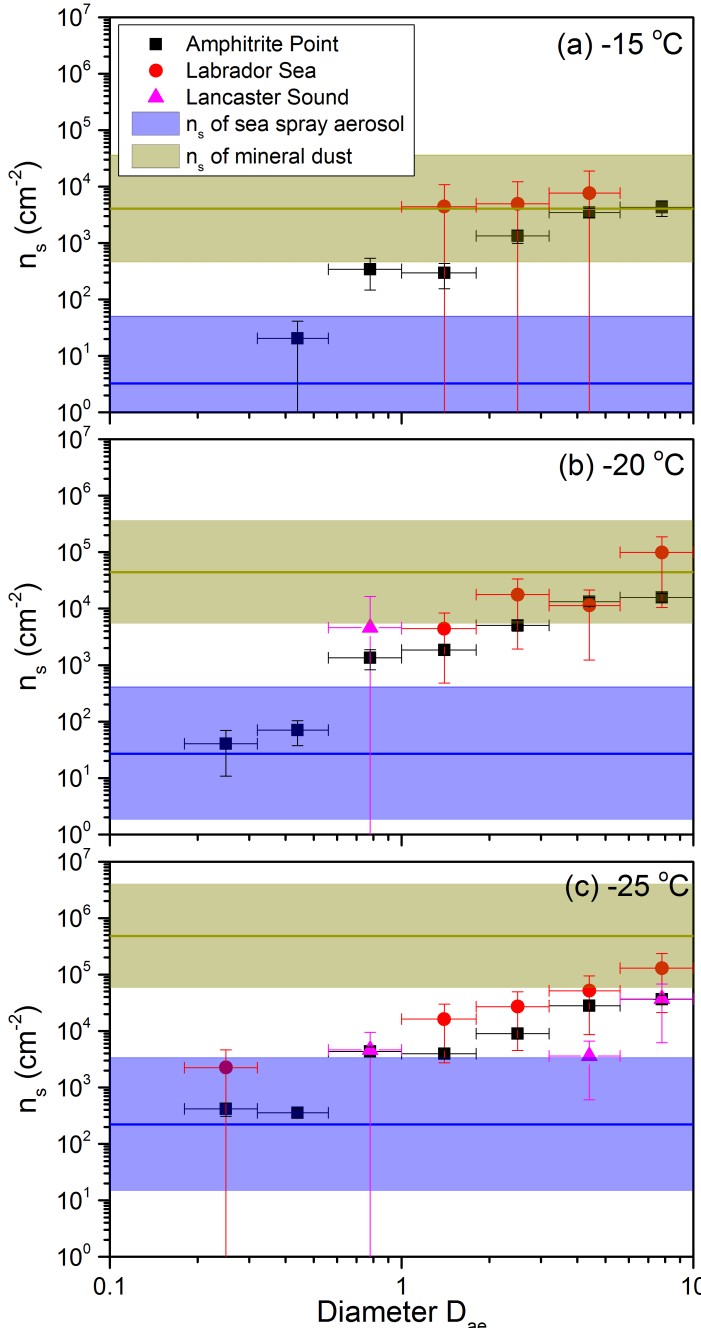

**Figure 7. Comparison of measured $n_s$ values with previously reported $n_s$ values of sea spray aerosol and mineral dust at -15 ºC, -20 ºC, and -25 ºC, respectively. The $n_s$ values of sea spray aerosol were taken from DeMott et al. (2016), and the $n_s$ values of mineral dust were taken from Niemand et al. (2012). The horizontal lines represent the calculated $n_s$ values from linear regression, and the coloured bands represent the 95 % prediction bands (see Fig. S5-S6 in the Supplement). Blue represents sea spray aerosol, and light green represents mineral dust.**

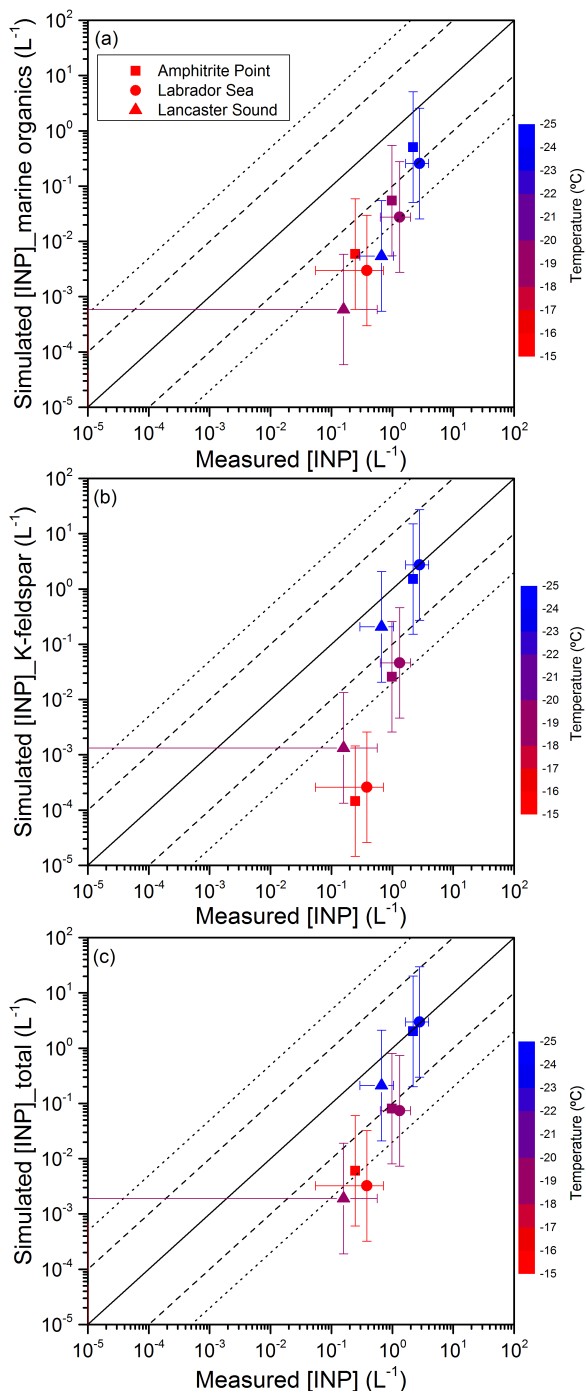

**Figure 8. Comparison of measured INP concentrations and (a) simulated INP concentrations from marine organics, (b) simulated INP concentrations from K-feldspar, and (c) simulated INP concentrations from both. The solid lines represent 1:1 ratio, the**

dashed and dotted lines represent one order and 1.5 orders of magnitude difference, respectively. The temperature is shown using a color scale. The simulated INP concentrations for Amphitrite Point, Labrador Sea, and Lancaster Sound correspond to mean concentrations for the months of August, July, and July, respectively. The uncertainties in the simulated concentrations are estimated to be around one order of magnitude based on the parameterization and model uncertainty (Harrison et al., 2016; Wilson et al., 2015).

