# Peer review of "Ice-nucleating ability of aerosol particles and possible sources at three coastal marine sites"

_Atmospheric Chemistry and Physics, 2018_

## Referee Comment (RC1) · Anonymous Referee #2 · 2 Apr 2018

Si et al. present a comprehensive observational and modeling study evaluating size-resolved INPs at multiple coastal locations. They found a relationship between particle diameter and fraction of INPs, indicating the larger particles were more efficient ice nucleators. Size-resolved ice nucleation studies such as this are needed to better characterize INP sources. Although this study provides valuable insight into INPs, I have outlined a few issues below that should be addressed prior to publication.

**General comments:**

Drying the sample flow to 2% seems quite extreme and is far below the GAW standard of 40% for the SMPS. Can the authors comment on how this dry of a sample flow may affect the ambient aerosol? I would assume these sort of conditions would remove semi-volatile species from the aerosol in addition to water, especially at these sizes. Although the authors do describe the corrections to the different diameter types and hygroscopic growth, the very large discrepancy between the APS and SMPS sampling conditions might not make them directly comparable given the possibility of other semi-volatile species that may have been removed.

I realize $n_s$ has been commonly used to represent INP data, but how representative is $n_s$ of the actual INP surface sites? The equation takes into account the surface area of *all* aerosols within a given size range, but if only 1 in $10^6$ particles are INPs as the authors define for 0.2 um particles, is $n_s$ realistic for the INP fraction? The authors should discuss any potential biases. Also, how was a definite size of INPs determined, given the MOUDI measures size ranges? In this case, shouldn't the aerosol surface area be defined by the same range of sizes from the SMPS and APS?

There seems to be disagreement between the air mass sources (especially at Amphitrite Point) and the source apportionment results (i.e., Fig 7). Can the authors comment on why the INPs appear to be of a more terrestrial origin yet air masses were predominantly from over the ocean? What sort of very localized sources could influence the samples?

**Minor comments:**

P2 l 39-43: The -35 C statement is redundant from the sentence above. Also, this statement should be reworded since INPs can initiate ice formation below -35 C (e.g., glassy organics, soot, sea salt).

P2 l 44-45: Please provide a reference for this statement.

P4, l104: Which 2 stages were analyzed?

P4, l125: How many droplets? What was the spacing? Were any neighboring droplet freezing effects apparent? For example, if droplets are too close, they can induce freezing in neighboring droplets.

P7, l188 and P8 l 222-223: Was there any issues with artifacts from storing the dishes at room temperature as opposed to freezing the samples? Also, could the authors comment on how there could be issues comparing samples from the different locations given the different storage conditions and duration?

P9, l264 on: Since the measurements were conducted at coastal locations, there is a likelihood that terrestrial sources of INPs may also influence the air sampled, especially given air mass trajectories show not all air masses originated from over the ocean. Can the authors comment on how this possible interference may have been dealt with, aside from the brief statement on the end of section 3.1?

P11, l 313: These concentrations seem fairly high for an Arctic marine atmosphere. What was the error or standard deviation of these averages? Were they just from when air masses originated over the ocean? Was new particle formation observed?

P12, l 349: Please provide equation for $n_s$.

P14, l 409: How was "marine biological activities" defined?

P14, l411: But air masses originated from over the ocean 94% of the time, so how would terrestrial sources be a dominant source of INPs? There seems to be some inconsistency between air mass sources in this manuscript as compared to the results from Mason et al. (2015a).

Figure 2: Given the MBL can often be quite low, especially in the Arctic, the color scale should be adjusted so that the 0 – 600 m range is easier to differentiate in the figure.

---

## Referee Comment (RC2) · Anonymous Referee #1 · 2 Apr 2018

Review of "Ice-nucleating efficiency of aerosol particles and possible sources at three coastal marine sites" by Si et al., submitted to ACPD:

The study described in this manuscript is an interesting addition to similar work by the group in which M. Si is working. It interprets measurements of atmospheric aerosol wrt. concentrations of ice nucleation particles (INP) and their possible sources, as well as effects of particle size on INP activity. The work is interesting and timely. However, I have a few major comments (besides for a number of smaller ones) that need to be addressed before the work can be published. The major comments mainly concern the amount of data (which is rather low) and a possible malfunctioning of one of the size spectrometers used and related consequences on the results.

But altogether, the study merits publication once again my comments will have been

considered and changes will have been implemented adequately.

Major comments:

The first comment concerns the amount of data used for the study. For two locations, only one measurement was made, while for the third location, the data used has already been used in a different study on INP before. This is properly stated in the text. And the results obtained herein evaluate these data in a new way, yielding more results. But the abstract had raised high expectations, and I was quite disappointed when I realized that the abundance of data included in this study is rather low. It should be made clear already in the abstract and again in the conclusions that the data base is not very strong. This might also influence the results, as one measurement does not deliver good statistics, and this has to be dealt with offensively and should be discussed.

I am also concerned about the misfit in the particle number size distributions between those number concentrations measured by the SMPS and by the APS. There is a large gap at $\sim$ 500nm which implies that one of the two instruments might have not worked well. As the majority of the particles is in the SMPS size range, as usual, and as these seem to fit well with literature (at least that is what is said on page 8, line 14), it might have been the APS, measuring roughly one order of magnitude too low concentrations (that's roughly the size of the gap, larger for Lancaster Sound, a little less for Amphitrite Point and only $\sim$*2 in the Labrador Sea). This would translate to the same magnitude of error (i.e., overestimation in this case) in INP concentrations and surface site densities, affecting many statements/results reported in the text. The observed strong increase in n_s for particles < 500nm to particles > 500nm likely is (at least in part) related to this gap between number concentrations as measured by SMPS and APS. Is there a way to find out what the problem might have been? Was there a total particle number counter deployed that could shed light onto this? In any case, this problem has to be critically discussed and related changes in the interpretation of the data has to be included in the revised version of the manuscript.

page 9, line 9: It is interesting that you find n_s depends on size. But why could that be? - The larger particles would have to consist of a material that is more ice active (per surface area) than the smaller ones. How should this come about? (A mechanism would have to be that e.g., clay minerals make smaller particles, and then, the larger the mineral dust particles get, the higher becomes their feldspar content, and isn't this unlikely?) - This observed increase could be a measurement bias (as mentioned above and again in my comment concerning Fig. 4). - This needs to be discussed!

page 6, line 30: Was the model really run for 2001? If yes, why not for the respective month of 2013 and 2014, i.e., when the sampling was done? Can it really be assumed that the average monthly INP concentration is the same every year? How much variation could there be expected, and where within this variation are your data? Could this influence your results?

Figure 4: Again, as this is where I first noticed it: There is a VERY pronounced gap between number concentrations measured with the SMPS and the APS. Do you have any explanation? This could potentially influence the derived n_s values and the fraction of INP per particle as well as your comparison with the model, particularly if it was the APS that did not operate well.

Minor and technical comments:

page 1, line 17: Add "particle" before "size".

chapter 2.1.2: Also mention the temperature and RH at which droplet formation was done (the text has to be complete, i.e., readers should not have to look this up in another paper)

equation (1): There seems to be a typo in the formular: $N_0$ should not be there as a factor. The value resulting from this equation has a wrong dimension (assuming the corrections factors are dimensionless, which is how they are given in the Supplement). Please check this carefully – see also e.g., Hader et al., 2014.

page 5, line 24: Reaching an RH below 2% by a silica gel diffusion dryer is quite low (even when the silica gel is changed every 24 hours), unless the dew point of the outside air is quite low, anyway. – Did you estimate this value or check it?

page 6, line 6: Please give wind-speed in SI units – knots is a unit many (including myself) may not be familiar with.

page 6, line 10: As the MOUDI was inside at least at Amphitrite Point, drying will automatically have occurred, as it will have been warmer in the container than it was outside. This typically leads to a quick drying. The way you formulated this here is correct, however, it may be good to hint at the fact that the RH will also not have been the outside one.

page 7, line 6 and page 10, line 12: The "-" is missing for Vergara-Temprado.

page 7, line 17-18: It might be better to formulate it in a different way. Basically you are saying that you don't know where the INP came from (marine or terrestrial and maybe even from further away), so please say something like: "Therefore it is not possible to determine if the INP are of marine or terrestrial origin and they may even have been long-range transported from sources more than three days away."

page 7, line 24: Check with chapter 2.3.1 – you give different dates for the sampling period at Amphitrite Point.

page 7, line 26: Clearly state how many samples you used for the present study.

page 8, line 14: Marine sites may differ depending on the sea spray production typical for the area. Were the two studies you cite here done at locations that are similar to yours in this regard? Please mention in the text if they are.

page 8, line 25 ff: This effect was already reported by Mertes et al. (2007), which therefore should be cited here or in the following paragraph.

page 9, line 26-27: Niemand et al. (2012) report n_s for dust samples in which all

particles were dust. But your way to determine n_s relates the ice activity to the total particle number concentrations that were measured, hence, this is not the exact same parameter. This needs to be clearly stated here. BTW: In DeMott et al. (2016), INP concentrations for an assumed total particle number concentration of 150 cm-3 were reported (at least that's the value the laboratory data are normalized to - this is said in the caption of Fig. 1 to which you are referring), which is a factor of 2 to 4 below your values – this should also at least be mentioned, although, admittedly, this will not change your results.

page 9, line 29-30: It should be motivated a bit stronger why you make these statements here. My comment above this one might be one reason, but I am sure you had more in mind when writing these lines.

figure 1: The insets of the pictures of the ship and measurement container might not be visible any more in a printed version, so if you care for them, you might want to make them larger (there is enough "empty" space on the map).

chapter S1: Add values for the growth factors used (and / or for values for kappa).

figure S3: In the caption you say that "Each data point was calculated by adding together the numbers from Fig. 4." Did you really simply add the data points? Looking at the number, this does not seem to have been the case, and it would have been totally wrong.

Literature:

Hader, J. D., T. P. Wright, and M. D. Petters (2014), Contribution of pollen to atmospheric ice nuclei concentrations, Atmos. Chem. Phys., 14(11), 5433-5449, doi:10.5194/acp-14-5433-2014.

Mertes, S., B. Verheggen, S. Walter, P. Connolly, M. Ebert, J. Schneider, K. N. Bower, J. Cozic, S. Weinbruch, U. Baltensperger, and E. Weingartner (2007), Counterflow virtual impactor based collection of small ice particles in mixed-phase clouds for the physicochemical characterization of tropospheric ice nuclei: sampler description and first case study, Aerosol Sci. Technol., 41, 848-864.

---

## Referee Comment (RC3) · Anonymous Referee #3 · 30 Apr 2018

Review of "Ice-nucleating efficiency of aerosol particles and possible sources at three coastal marine sites"

Si et al. (2018) investigate sources of ice nucleating particles (INPs) from three coastal sites with a combined measurement-modeling approach. Measurements were taken with a suite of well-established instrumentation, that allowed quantification of INPs by an active site density function $n_s$. The results were compared with the output of a global INP model, and it was found that the two INP model of K-feldpsar and marine organics missed a high temperature INP source. Speculation as to what this source is was carried out reasonably. The paper is well written and the figures are clear. I think this study merits publication in ACP after some minor concerns are addressed.

The first pertains to the global INP model. I would appreciate the inclusion of a more critical account of the limitations of the model when being compared to ground based measurements. There is a big jump in the conclusion that there is a missing source of INP. For example, how can the authors be sure the measurements aren't artificially inflating the INP activity at higher temperatures by sampling from the ground? The global INP model is supposed to shed light on what, statistically and on long/large enough scales, INPs matter. The measurements on the other hand are happening locally from boundary layer air. Please investigate this point further.

Other comments are specific to the text and are outlined below.

P2 L1-2: The studies cited do not conclude that INPs "significantly impact the frequencies, lifetime, and optical properties of ice and mixed-phase clouds". Consider changing to something less assertive like "may impact".

P9 L5: $n_s$ as a function of size is a useful approach here. However, there are issues with surface area corrections that make $n_s$ not without shortcomings. Studies by Beydoun et al. (2016), Emersic et al. (2015), and Hiranuma et al. (2015) discuss these shortcoming and should be included in an additional discussion on what kind of limitations the authors expect when analyzing $n_s$ against surface area.

P11 L5: The authors can do a better job here of synthesizing their results and suggesting a way forward. For example, on the measurement side, samples can be investigated with a chemical composition analysis. On the modelling side, large eddy simulations can discern whether boundary layer INP are different than free atmospheric INPs simulated by the global model. So I think there's a bit more room here for discussing future efforts.

Technical correction:

$n_s$ is a surface area density, not an efficiency. It has units of m$^{-2}$ and does not range from 0 to 1 (like an efficiency would). You may also want to consider changing that in the title as well. Please refer to Vali et al. (2014) to ensure INP specific terminology is consistent.

References

Beydoun, H., Polen, M., & Sullivan, R. C. (2016). Effect of particle surface area on ice active site densities retrieved from droplet freezing spectra. *Atmospheric Chemistry and Physics*, *16*(20), 13359–13378. https://doi.org/10.5194/acp-16-13359-2016

Emersic, C., Connolly, P. J., Boult, S., Campana, M., & Li, Z. (2015). Investigating the discrepancy between wet suspension and dry-dispersion derived ice nucleation efficiency of mineral particles. *Atmospheric Chemistry and Physics*, *15*(19), 11311–11326. https://doi.org/10.5194/acp-15-11311-2015

Hiranuma, N., Augustin-Bauditz, S., Bingemer, H., Budke, C., Curtius, J., Danielczok, A., et al. (2015). A comprehensive laboratory study on the immersion freezing behavior of illite NX particles: a comparison of 17 ice nucleation measurement techniques. *Atmospheric Chemistry and Physics*, *15*(5), 2489–2518. https://doi.org/10.5194/acp-15-2489-2015

Vali, G., DeMott, P., Möhler, O., & Whale, T. F. (2014). Ice nucleation terminology. *Atmospheric Chemistry and Physics Discussions*, *14*, 22155–22162. https://doi.org/10.5194/acpd-14-22155-2014

---

## Author Comment (AC1) · 12 Jun 2018

The comment was uploaded in the form of a supplement:
https://www.atmos-chem-phys-discuss.net/acp-2018-81/acp-2018-81-AC1-
supplement.pdf
* * *

---

## Author Response (AR1)

Review of "Ice-nucleating efficiency of aerosol particles and possible sources at three coastal marine sites" by Si et al., submitted to ACPD:

The study described in this manuscript is an interesting addition to similar work by the group in which M. Si is working. It interprets measurements of atmospheric aerosol wrt. concentrations of ice nucleation particles (INP) and their possible sources, as well as effects of particle size on INP activity. The work is interesting and timely. However, I have a few major comments (besides for a number of smaller ones) that need to be addressed before the work can be published. The major comments mainly concern the amount of data (which is rather low) and a possible malfunctioning of one of the size spectrometers used and related consequences on the results.

But altogether, the study merits publication once again my comments will have been considered and changes will have been implemented adequately.

Major comments:

*[1]* The first comment concerns the amount of data used for the study. For two locations, only one measurement was made, while for the third location, the data used has already been used in a different study on INP before. This is properly stated in the text. And the results obtained herein evaluate these data in a new way, yielding more results. But the

abstract had raised high expectations, and I was quite disappointed when I realized that the abundance of data included in this study is rather low. It should be made clear already in the abstract and again in the conclusions that the data base is not very strong. This might also influence the results, as one measurement does not deliver good statistics, and this has to be dealt with offensively and should be discussed.

*[A1]* To address the referee's comment, in the Abstract and Summary and conclusions we have added the following sentence:

"For Amphitrite Point, 23 sets of samples were analyzed, and for Labrador Sea and Lancaster Sound, one set of samples was analyzed for each location".

In addition, we have added the following sentence in the Summary and conclusions:

"Since only one sample was analyzed for both Labrador Sea and Lancaster Sound, additional samples should be collected and analyzed at these locations to determine the general applicability of the results presented here for these locations".

*[2]* I am also concerned about the misfit in the particle number size distributions between those number concentrations measured by the SMPS and by the APS. There is a large gap at ~ 500nm which implies that one of the two instruments might have not worked well. As the majority of the particles is in the SMPS size range, as usual, and as these seem to fit well with literature (at least that is what is said on page 8, line 14), it might have been the APS, measuring roughly one order of magnitude too low concentrations (that's roughly the size of the gap, larger for Lancaster Sound, a little less for Amphitrite Point and only ~*2 in the Labrador Sea). This would translate to the same magnitude of error (i.e., overestimation in this case) in INP concentrations and surface site densities, affecting many statements/results reported in the text. The observed strong increase in n_s for particles < 500nm to particles > 500nm likely is (at least in part) related to this gap between number concentrations as measured by SMPS and APS. Is there a way to find out what the problem might have been? Was there a total particle number counter deployed that could shed light onto this? In any case, this problem has to be critically discussed and related changes in the interpretation of the data has to be included in the revised version of the manuscript.

*[A2]* For short periods of time during the Amphitrite Point campaign, there was an additional SMPS measurement and an additional APS measurement. During these short periods of time, the total number concentrations measured by the two SMPS instruments agreed within 10 % for the size range relevant for this paper and where overlap occurred (0.18-0.3 μm). In addition, the total number concentrations measured by the two APS measurements agreed within 10 % for the size range of 0.7-10 μm. This agreement is within the uncertainty of the instruments.

After going back and investigating the size distribution data used in this study in more detail, we conclude that the gap between the SMPS and APS data is most likely due to (1) a drop off in the efficiency of the APS at size channels below 0.7 μm, and (2) an uncertainty in the hygroscopic properties at Amphitrite Point used to correct

the SMPS data for hygroscopic growth. Regarding (1), drop off in the efficiency of an APS at sizes below 0.7 µm has often been observed previously (Beddows et al., 2010). To address this issue, in the revised manuscript, the APS data at sizes below 0.7 µm has been omitted, as done previously (Maguhn et al., 2003). Regarding (2), in the revised manuscript, we have used a different method to correct the SMPS data for hygroscopic growth at Amphitrite Point. Specifically, we used a free parameter to correct for the hygroscopic growth, which resulted in the optimal overlap between the SMPS and APS data at Amphitrite Point. This type of approach has been used successfully in the past to merge SMPS and APS data (Beddows et al., 2010). After limiting the APS data to sizes of 0.7-10 µm and using a free parameter to correct for the hygroscopic growth at Amphitrite Point, the misfit in the particle number size distribution is much less.

In addition, we have included a comparison between the size distributions measured at all three sites with the size distributions measured previously at a Mid-latitude North-Atlantic marine boundary layer site by O'Dowd et al. (2001). Please see changes in Section 2.4 and Fig. S3 in the revised Supplement.

*[3]* page 9, line 9: It is interesting that you find n_s depends on size. But why could that be? - The larger particles would have to consist of a material that is more ice active (per surface area) than the smaller ones. How should this come about? (A mechanism would have to be that e.g., clay minerals make smaller particles, and then, the larger the mineral dust particles get, the higher becomes their feldspar content, and isn't this unlikely?) - This observed increase could be a measurement bias (as mentioned above and again in my comment concerning Fig. 4). - This needs to be discussed!

*[A3]* The size distribution of $n_s$ can be qualitatively explained by considering four different types of aerosol particles each having progressively larger geometric mean diameters and $n_s$ values. As an example, consider a mixture of: a) sulfate aerosols internally mixed with black carbon with a small $n_s$ and small geometric mean diameter, b) sea salt aerosols with a larger $n_s$ and larger geometric mean diameter, c) clay particles with a larger $n_s$ and larger geometric mean diameter, and d) biological particles from terrestrial sources with the largest $n_s$ and largest geometric mean diameter. To address the referee's comments, this information has been added to the Summary and conclusions in the revised manuscript.

*[4]* page 6, line 30: Was the model really run for 2001? If yes, why not for the respective month of 2013 and 2014, i.e., when the sampling was done? Can it really be assumed that the average monthly INP concentration is the same every year? How much variation could there be expected, and where within this variation are your data? Could this influence your results?

*[A4]* Model data from the year 2001 was used because this model output was available from previous studies. This has been made clear in the revised manuscript. In addition, to address the referee's comments, the following information has been added to Section 2.6 of the revised manuscript:

"As mentioned above and as done previously, the model output from the year 2001 was compared with measurements from different years. The inter-annual variability of aerosol concentrations simulated in the model is expected to be up to a factor of 2 due to differences in meteorological conditions (Marmer and Langmann, 2007). Model output for the year 2001 has been found to be able to reproduce the mass concentrations of mineral dust and marine organic aerosols within an order of magnitude with observations made in various years (Vergara-Temprado et al., 2017). Furthermore, the model output for the year 2001 was able to reproduce 62 % of the INP concentrations measured from studies spanning from 1973 to 2016 within an order of magnitude, which is the uncertainty in the predicted INP concentrations reported here (Fig. 8)."

*[5]* Figure 4: Again, as this is where I first noticed it: There is a VERY pronounced gap between number concentrations measured with the SMPS and the APS. Do you have any explanation? This could potentially influence the derived n_s values and the fraction of INP per particle as well as your comparison with the model, particularly if it was the APS that did not operate well.

*[A5]* Please see *[A2]* above.

Minor and technical comments:

*[6]* page 1, line 17: Add "particle" before "size".

*[A6]* The word "particle" has been added before "size".

*[7]* chapter 2.1.2: Also mention the temperature and RH at which droplet formation was done (the text has to be complete, i.e., readers should not have to look this up in another paper)

*[A7]* To address the referee's comment, the following sentence has been added to Section 2.1.2:

"The temperature was decreased to approximately 0 ºC, and the relative humidity was increased to above water saturation using a humidified flow of He (99.999 %, Praxair), resulting in the condensation and growth of water droplets on the collected particles."

*[8]* equation (1): There seems to be a typo in the formular: N_0 should not be there as a factor. The value resulting from this equation has a wrong dimension (assuming the corrections factors are dimensionless, which is how they are given in the Supplement). Please check this carefully – see also e.g., Hader et al., 2014.

*[A8]* Thanks for checking the equation in our manuscript. The units used in Equation (1) are correct and consistent with Hader et al. (2014). To address the referee's comments, in the revised manuscript Equation (1) in Section 2.1.2 has been separated into two equations. In this case, the consistency between our calculations and the equation in Hader et al. (2014) should be more obvious.

*[9]* page 5, line 24: Reaching an RH below 2% by a silica gel diffusion dryer is quite low (even when the silica gel is changed every 24 hours), unless the dew point of the outside air is quite low, anyway. – Did you estimate this value or check it?

> *[A9]* This value was not measured on site, but has been checked in the lab with the same technique. To address the referee's comment, the following has been added to Section 2.3.1 of the revised manuscript:
>
> "Three successive diffusion dryers were used prior to sampling with the SMPS, and the silica was exchanged and dried in an oven every 24 h. Although not measured on site in this campaign, this technique has been found to always reduce the RH to less than 20 %, and usually to less than 2 % (Ladino et al., 2014; Yakobi-Hancock et al., 2014)."

*[10]* page 6, line 6: Please give wind-speed in SI units – knots is a unit many (including myself) may not be familiar with.

> *[A10]* The unit has been changed to SI unit (km h$^{-1}$).

*[11]* page 6, line 10: As the MOUDI was inside at least at Amphitrite Point, drying will automatically have occurred, as it will have been warmer in the container than it was outside. This typically leads to a quick drying. The way you formulated this here is correct, however, it may be good to hint at the fact that the RH will also not have been the outside one.

> *[A11]* The drying effect due to the warmer temperature inside the container at Amphitrite Point was discussed in Section 2.3.1. Does the referee want us to repeat this information again in Section 2.4? Sorry, this was not clear to us.

*[12]* page 7, line 6 and page 10, line 12: The "-" is missing for Vergara-Temprado.

> *[A12]* Thanks for pointing out this mistake. It has been corrected in the revised manuscript.

*[13]* page 7, line 17-18: It might be better to formulate it in a different way. Basically you are saying that you don't know where the INP came from (marine or terrestrial and maybe even from further away), so please say something like: "Therefore it is not possible to determine if the INP are of marine or terrestrial origin and they may even have been long-range transported from sources more than three days away."

> *[A13]* Thanks for the advice. The sentence has been rephrased as following:
>
> "Therefore, it was not possible to determine if the INPs are of marine or terrestrial origin based on the back trajectories alone. INPs may even have been long-range transported from sources that were not reached by the 3-day back trajectories".

*[14]* page 7, line 24: Check with chapter 2.3.1 – you give different dates for the sampling period at Amphitrite Point.

*[A14]* Thanks for pointing out this mistake. The date in chapter 2.3.1 has been corrected.

*[15]* page 7, line 26: Clearly state how many samples you used for the present study.

*[A15]* To address the referee's comment, the number of samples has been clearly stated in Section 3.2 of the revised manuscript.

*[16]* page 8, line 14: Marine sites may differ depending on the sea spray production typical for the area. Were the two studies you cite here done at locations that are similar to yours in this regard? Please mention in the text if they are.

*[A16]* To address the referee's comment, in the revised manuscript we have added the location of the study referenced. In addition, a figure comparing the size distributions from the current study and the study referenced has been added to the Supplement (Fig. S3). Finally, additional details of the study referenced have been included in the Supplement (figure caption of Fig. S3).

*[17]* page 8, line 25 ff: This effect was already reported by Mertes et al. (2007), which therefore should be cited here or in the following paragraph.

*[A17]* Thanks for pointing this out. The result from Mertes et al. (2007) has been cited in Section 3.4 of the revised manuscript.

*[18]* page 9, line 26-27: Niemand et al. (2012) report n_s for dust samples in which all particles were dust. But your way to determine n_s relates the ice activity to the total particle number concentrations that were measured, hence, this is not the exact same parameter. This needs to be clearly stated here. BTW: In DeMott et al. (2016), INP concentrations for an assumed total particle number concentration of 150 cm-3 were reported (at least that's the value the laboratory data are normalized to - this is said in the caption of Fig. 1 to which you are referring), which is a factor of 2 to 4 below your values – this should also at least be mentioned, although, admittedly, this will not change your results.

*[A18]* To address the referee's comment, we have added the following statement to Section 3.5:

"Note, the data from Niemand et al. (2012) corresponds to the $n_s$ values of only mineral dust particles, whereas the $n_s$ values reported here correspond to the total aerosol particles, as mentioned above".

Regarding the normalization factor in DeMott et al. (2016), this is certainly relevant when reporting the concentrations of INPs (Fig. 1 in DeMott et al., 2016), but we do not think this is relevant when discussing the $n_s$ values (Fig. 3 in DeMott et al., 2016), which is the focus of our manuscript. In other words, scaling to 150 cm$^{-3}$ was not used when calculating $n_s$. Please let us know if we misunderstood the referee's comment.

*[19]* page 9, line 29-30: It should be motivated a bit stronger why you make these statements here. My comment above this one might be one reason, but I am sure you had more in mind when writing these lines.

*[A19]* We hope our response to *[18]* provided a stronger motivation.

*[20]* figure 1: The insets of the pictures of the ship and measurement container might not be visible any more in a printed version, so if you care for them, you might want to make them larger (there is enough "empty" space on the map).

*[A20]* The insets in Fig. 1 have been made larger to be more visible.

*[21]* chapter S1: Add values for the growth factors used (and / or for values for kappa).

*[A21]* In the revised Supplement, we have used a different method to correct for hygroscopic growth (see *[A2]* above).

*[22]* figure S3: In the caption you say that "Each data point was calculated by adding together the numbers from Fig. 4." Did you really simply add the data points? Looking at the number, this does not seem to have been the case, and it would have been totally wrong.

*[A22]* Each data point was calculated by averaging the numbers in each size bin from Fig. 4. This has been corrected in the caption of Fig. S4 of the revised Supplement.

Literature:

Hader, J. D., T. P. Wright, and M. D. Petters (2014), Contribution of pollen to atmospheric ice nuclei concentrations, Atmos. Chem. Phys., 14(11), 5433-5449, doi:10.5194/acp-14-5433-2014.

Mertes, S., B. Verheggen, S. Walter, P. Connolly, M. Ebert, J. Schneider, K. N. Bower, J. Cozic, S. Weinbruch, U. Baltensperger, and E. Weingartner (2007), Counterflow virtual impactor based collection of small ice particles in mixed-phase clouds for the physico-chemical characterization of tropospheric ice nuclei: sampler description and first case study, Aerosol Sci. Technol., 41, 848-864.

**Anonymous Referee #2**

Si et al. present a comprehensive observational and modeling study evaluating size-resolved INPs at multiple coastal locations. They found a relationship between particle diameter and fraction of INPs, indicating the larger particles were more efficient ice nucleators. Size-resolved ice nucleation studies such as this are needed to better characterize INP sources. Although this study provides valuable insight into INPs, I have outlined a few issues below that should be addressed prior to publication.

**General comments:**

*[23]* Drying the sample flow to 2% seems quite extreme and is far below the GAW standard of 40% for the SMPS. Can the authors comment on how this dry of a sample flow may affect the ambient aerosol? I would assume these sort of conditions would remove semi-volatile species from the aerosol in addition to water, especially at these sizes. Although the authors do describe the corrections to the different diameter types and hygroscopic growth, the very large discrepancy between the APS and SMPS sampling conditions might not make them directly comparable given the possibility of other semi-volatile species that may have been removed.

> *[A23]* To address the referee's comment, the following has been added to Section 2.3.1:
>
> "For typical atmospheric conditions, the equilibration timescale for gas-particle partitioning of semivolatile organic species is on the order of minutes to tens of minutes (Saleh et al., 2013). In contrast, the residence time in the dryers during sampling in the current study was approximately 10s. Therefore, removal of semi-volatile organic species during drying may not have been a large issue but cannot be completely ruled out."

*[24]* I realize $n_S$ has been commonly used to represent INP data, but how representative is $n_S$ of the actual INP surface sites? The equation takes into account the surface area of *all* aerosols within a given size range, but if only 1 in $10^6$ particles are INPs as the authors define for 0.2 um particles, is $n_S$ realistic for the INP fraction? The authors should discuss any potential biases. Also, how was a definite size of INPs determined, given the MOUDI measures size ranges? In this case, shouldn't the aerosol surface area be defined by the same range of sizes from the SMPS and APS?

> *[A24]* The $n_s$ values reported here were calculated with the total surface area of the aerosol. In this case, the $n_s$ values correspond to a lower limit of the $n_s$ values of the INPs. To address the referee's comment, in Section 3.5 we have added the following sentence:
>
> "Since this equation considers the surface area of all aerosol particles, rather than the surface area of just the INPs, the calculated $n_s$ values are lower limits to the $n_s$ values for the INPs".

> Regarding the second comment, yes, the sizes of the INPs were determined from the MOUDI size ranges. The aerosol surface area is indeed calculated by integrating the surface area measured by the SMPS and APS within the same size range as INPs.

*[25]* There seems to be disagreement between the air mass sources (especially at Amphitrite Point) and the source apportionment results (i.e., Fig 7). Can the authors comment on why the INPs appear to be of a more terrestrial origin yet air masses were predominantly from over the ocean? What sort of very localized sources could influence the samples?

> *[A25]* INPs of terrestrial origin could come from local vegetation or from long-range transport. Local vegetation could potentially release enough INPs to overwhelm a small INP source from the ocean. Long-range transported desert dust may still dominate the INP population after several days of transport over the ocean (Vergara-Temprado et al., 2017). To address the referee's comment, the following text has been added to Section 3.1 of the revised manuscript:

> "At Amphitrite Point, although the air masses were predominantly from the ocean based on the back trajectory analysis, the air masses did pass over local vegetation including western coastal hemlock. This local vegetation could potentially release enough INPs to overwhelm a small INP source from the ocean. Therefore, it was not possible to determine if the INPs are of marine or terrestrial origin based on the back trajectories alone. INPs may even have been long-range transported from sources that were not reached by the 3-day back trajectories (Vergara-Temprado et al., 2017)."

**Minor comments:**

*[26]* P2 l 39-43 (Page 1, line 28-30 in the ACPD version): The -35 C statement is redundant from the sentence above. Also, this statement should be reworded since INPs can initiate ice formation below -35 C (e.g., glassy organics, soot, sea salt).

> *[A26]* The sentences in the Introduction have been reworded in the revised manuscript as following:

> "Aerosol particles are ubiquitous in the atmosphere, yet only a small fraction of these particles, referred to as ice nucleating particles (INPs), are able to initiate the formation of ice at temperatures warmer than homogeneous freezing temperatures. INPs may impact the frequencies, lifetime, and optical properties of ice and mixed-phase clouds (Andreae and Rosenfeld, 2008; Cziczo and Abbatt, 2001; Lohmann and Feichter, 2005)".

*[27]* P2 l 44-45 (Page 2, line 3 in the ACPD version): Please provide a reference for this statement.

> *[A27]* A reference has been added to the revised manuscript.

*[28]* P4, l104 (Page 3, line 12 in the ACPD version): Which 2 stages were analyzed?

*[A28]*. Stages 2 through 8 were analyzed (seven stages in total). This sentence has been reworded as following to make this point clear in the revised manuscript:

"Stages 2 through 8 of the MOUDI were analyzed for this study (seven stages in total)".

*[29]* P4, l125 (Page 3, line 27 in the ACPD version): How many droplets? What was the spacing? Were any neighboring droplet freezing effects apparent? For example, if droplets are too close, they can induce freezing in neighboring droplets.

*[A29]* On average, approximately 40 droplets were analyzed in each experiment. The spacing between droplets varied, but was roughly 100 μm on average. Freezing of neighboring droplets was apparent in some cases, and this was taken into account while calculating the INP concentrations. This information has been added to Section 2.1.2 of the revised manuscript as following:

"On average, approximately 40 droplets were analyzed in each experiment. The final droplet size was approximately 50-150 μm in diameter, and the spacing between droplets was roughly 100 μm, on average. After the formation of droplets, the flow cell was cooled down to -40 ℃ at a rate of -10 ℃ min$^{-1}$ while images of the droplets were recorded. During this process, most freezing events occurred by immersion freezing, while approximately 10 % occurred by contact freezing, which refers to the freezing of liquid droplets caused by contact with neighbouring frozen droplets. When calculating INP concentrations, the contact freezing was accounted for in two ways: (i) an upper limit to the fraction frozen by immersion freezing was calculated by assuming all the contact freezing droplets froze by immersion freezing; (ii) a lower limit to the fraction frozen by immersion freezing was calculated by assuming all the contact freezing droplets remained liquid until the homogeneous freezing temperature was reached".

*[30]* P7, l188 and P8 l 222-223 (Page 5, line 14 and page 6, line 7 in the ACPD version): Was there any issues with artifacts from storing the dishes at room temperature as opposed to freezing the samples? Also, could the authors comment on how there could be issues comparing samples from the different locations given the different storage conditions and duration?

*[A30]* Samples were stored under dry conditions and at either room temperature or 4 ℃. Samples were not frozen. To address the referee's comments, in the revised manuscript, we have pointed out that additional studies are needed to determine the effect of storage on INP concentrations. Specifically, the following has been added to Section 2.3.2:

"In contrast, the samples collected at Amphitrite Point were stored at room temperature and relative humidity for less than 24 h prior to INP analysis, as mentioned above. Studies are needed to determine the effect of sample storage conditions on measured INP concentrations".

*[31]* P9, l264 on: Since the measurements were conducted at coastal locations, there is a

likelihood that terrestrial sources of INPs may also influence the air sampled, especially given air mass trajectories show not all air masses originated from over the ocean. Can the authors comment on how this possible interference may have been dealt with, aside from the brief statement on the end of section 3.1?

*[A31]* We fully agree with the referee that terrestrial sources of INPs may also influence the air sampled. Both the comparison with literature $n_s$ values and the comparison with simulated INP concentrations addressed the importance of terrestrial sources to the INP population. To further address the referee's comment, we have expanded the discussion in Section 3.1 on possible terrestrial sources of INP. Specifically, the following has been added:

"At Amphitrite Point, although the air masses were predominantly from the ocean based on the back trajectory analysis, the air masses did pass over local vegetation including coastal western hemlock. This local vegetation could release enough INPs to overwhelm a small INP source from the ocean. Therefore, it was not possible to determine if the INPs are of marine or terrestrial origin based on the back trajectories alone. INPs may even have been long-range transported from sources that were not reached by the 3-day back trajectories".

*[32]* P11, l 313 (Page 8, line 11 in the ACPD version): These concentrations seem fairly high for an Arctic marine atmosphere. What was the error or standard deviation of these averages? Were they just from when air masses originated over the ocean? Was new particle formation observed?

*[A32]* To address the referee's comments, the standard deviation of the average total number concentration has been added to the revised manuscript. The majority of the air masses at Lancaster Sound were from over the ocean (63 % of the time), though the air masses also passed over the land. The air mass sources were discussed in Section 3.1. New particle formation has been observed in the Canadian Arctic marine boundary layer during the summer (Burkart et al., 2017). This information has been added to Section 3.3 of the revised manuscript.

*[33]* P12, l 349 (Page 9, line 6 in the ACPD version): Please provide equation for $n_S$.

*[A33]* An equation for $n_s$ has been added as Eq. (3) to Section 3.5 of the revised manuscript.

*[34]* P14, l 409 (Page 10, line 19 in the ACPD version): How was "marine biological activities" defined?

*[A34]* In Mason et al. (2015a), methanesulfonic acid (MSA) was used as a tracer of marine biological activities. This information has been added to Section 3.6 of the revised manuscript.

*[35]* P14, l411 (Page 10, line 20 in the ACPD version): But air masses originated from over the ocean 94% of the time, so how would terrestrial sources be a dominant source of INPs? There seems to be some inconsistency between air mass sources in this manuscript

as compared to the results from Mason et al. (2015a).

*[A35]* See *[A31]* above.

*[36]* Figure 2: Given the MBL can often be quite low, especially in the Arctic, the color scale should be adjusted so that the 0 – 600 m range is easier to differentiate in the figure.

*[A36]* The color scale in Figure 2 has been changed to a log scale so that the 0-600 m range is easier to differentiate.

**Anonymous Referee #3**

Review of "Ice-nucleating efficiency of aerosol particles and possible sources at three coastal marine sites"

Si et al. (2018) investigate sources of ice nucleating particles (INPs) from three coastal sites with a combined measurement-modeling approach. Measurements were taken with a suite of well- established instrumentation, that allowed quantification of INPs by an active site density function $n_S$. The results were compared with the output of a global INP model, and it was found that the two INP model of K-feldpsar and marine organics missed a high temperature INP source. Speculation as to what this source is was carried out reasonably. The paper is well written and the figures are clear. I think this study merits publication in ACP after some minor concerns are addressed.

*[37]* The first pertains to the global INP model. I would appreciate the inclusion of a more critical account of the limitations of the model when being compared to ground based measurements. There is a big jump in the conclusion that there is a missing source of INP. For example, how can the authors be sure the measurements aren't artificially inflating the INP activity at higher temperatures by sampling from the ground? The global INP model is supposed to shed light on what, statistically and on long/large enough scales, INPs matter. The measurements on the other hand are happening locally from boundary layer air. Please investigate this point further.

> *[A37]* The measurements were compared with predictions in the lowest level in the model. Since the model includes a parameterization of boundary layer turbulence (Holtslag and Boville, 1993), a comparison between measurements at the surface with predictions in the lowest level of the model should be appropriate. To address the referee's comments, this information has been included in Section 2.6 of the revised manuscript. In addition, we have also added the following caveat to the Summary and conclusions:
>
> "In addition, since the results presented here correspond to surface measurements, similar studies as a function of altitude are needed to determine if these results are applicable to higher altitudes and to the free troposphere."

Other comments are specific to the text and are outlined below.

*[38]* P2 L1-2: The studies cited do not conclude that INPs "significantly impact the frequencies, lifetime, and optical properties of ice and mixed-phase clouds". Consider changing to something less assertive like "may impact".

> *[A38]* Thank you for the suggestion. The wording has been changed to "may impact".

*[39]* P9 L5: $n_S$ as a function of size is a useful approach here. However, there are issues

with surface area corrections that make $n_s$ not without shortcomings. Studies by Beydoun et al. (2016), Emersic et al. (2015), and Hiranuma et al. (2015) discuss these shortcoming and should be included in an additional discussion on what kind of limitations the authors expect when analyzing $n_s$ against surface area.

*[A39]* The limitation of $n_s$ has been discussed in the revised manuscript. Specifically, the following has been added to Section 3.5 of the revised manuscript:

*"The surface active site density, $n_s$, represents the number of ice nucleation sites per surface area (Connolly et al., 2009; Hoose and Möhler, 2012; Vali et al., 2015). This parameterization assumes that freezing is independent of time and can be scaled with surface area. Although these assumptions may not be accurate in all cases (Beydoun et al., 2016; Emersic et al., 2015; Hiranuma et al., 2015), $n_s$ is commonly used to describe freezing data due, in part, to its simplicity".*

*[40]* P11 L5: The authors can do a better job here of synthesizing their results and suggesting a way forward. For example, on the measurement side, samples can be investigated with a chemical composition analysis. On the modelling side, large eddy simulations can discern whether boundary layer INP are different than free atmospheric INPs simulated by the global model. So I think there's a bit more room here for discussing future efforts.

*[A40]* To address the referee's comments, we have added the following text to the end of the Summary and conclusions:

"Since only one sample was analyzed for both Labrador Sea and Lancaster Sound, additional samples should be collected and analyzed at these locations to determine the general applicability of the results presented here for these locations. In addition, since the results presented here correspond to surface measurements, similar studies as a function of altitude are needed to determine if these results are applicable to higher altitudes and to the free troposphere. Comparison with predictions of INPs from a high-resolution model would also be useful to assess the importance of local INP sources. Studies of the chemical composition of the INPs are also needed to test the conclusions reached in the current study".

Technical correction:

*[41]* $n_s$ is a surface area density, not an efficiency. It has units of m$^{-2}$ and does not range from 0 to 1 (like an efficiency would). You may also want to consider changing that in the title as well. Please refer to Vali et al. (2014) to ensure INP specific terminology is consistent.

*[A41]* "Efficiency" has been changed to "ability" in the revised manuscript.

Literature list from referee #3

Beydoun, H., Polen, M., & Sullivan, R. C. (2016). Effect of particle surface area on ice active site densities retrieved from droplet freezing spectra. *Atmospheric Chemistry and Physics*, *16*(20), 13359–13378. https://doi.org/10.5194/acp-16-13359-2016

Emersic, C., Connolly, P. J., Boult, S., Campana, M., & Li, Z. (2015). Investigating the discrepancy between wet suspension and dry-dispersion derived ice nucleation efficiency of mineral particles. *Atmospheric Chemistry and Physics*, *15*(19), 11311–11326. https://doi.org/10.5194/acp-15-11311-2015

Hiranuma, N., Augustin-Bauditz, S., Bingemer, H., Budke, C., Curtius, J., Danielczok, A., et al. (2015). A comprehensive laboratory study on the immersion freezing behavior of illite NX particles: a comparison of 17 ice nucleation measurement techniques. *Atmospheric Chemistry and Physics*, *15*(5), 2489–2518. https://doi.org/10.5194/acp-15-2489-2015

Vali, G., DeMott, P., Möhler, O., & Whale, T. F. (2014). Ice nucleation terminology. *Atmospheric Chemistry and Physics Discussions*, *14*, 22155–22162. https://doi.org/10.5194/acpd-14-22155-2014

**References for author's response:**

Andreae, M. O. and Rosenfeld, D.: Aerosol-cloud-precipitation interactions. Part 1. The nature and sources of cloud-active aerosols, Earth-Science Rev., 89(1–2), 13–41, doi:10.1016/j.earscirev.2008.03.001, 2008.

Beddows, D. C. S., Dall'osto, M. and Harrison, R. M.: An Enhanced Procedure for the Merging of Atmospheric Particle Size Distribution Data Measured Using Electrical Mobility and Time-of-Flight Analysers, Aerosol Sci. Technol., 44(11), 930–938, doi:10.1080/02786826.2010.502159, 2010.

Beydoun, H., Polen, M. and Sullivan, R. C.: Effect of particle surface area on ice active site densities retrieved from droplet freezing spectra, Atmos. Chem. Phys., 16(20), 13359–13378, doi:10.5194/acp-16-13359-2016, 2016.

Burkart, J., Willis, M. D., Bozem, H., Thomas, J. L., Law, K., Hoor, P., Aliabadi, A. A., Köllner, F., Schneider, J., Herber, A., Abbatt, J. P. D. and Leaitch, W. R.: Summertime observations of elevated levels of ultrafine particles in the high Arctic marine boundary layer, Atmos. Chem. Phys., 17(8), 5515–5535, doi:10.5194/acp-17-5515-2017, 2017.

Connolly, P. J., Möhler, O., Field, P. R., Saathoff, H., Burgess, R., Choularton, T. and Gallagher, M.: Studies of heterogeneous freezing by three different desert dust samples, Atmos. Chem. Phys., 9(8), 2805–2824, doi:10.5194/acp-9-2805-2009, 2009.

Cziczo, D. J. and Abbatt, J. P. D.: Ice nucleation in NH4HSO4, NH4NO3, and H2SO4 aqueous particles: Implications for circus cloud formation, Geophys. Res. Lett., 28(6), 963–966, doi:10.1029/2000GL012568, 2001.

DeMott, P. J., Hill, T. C. J., McCluskey, C. S., Prather, K. A., Collins, D. B., Sullivan, R. C., Ruppel, M. J., Mason, R. H., Irish, V. E., Lee, T., Hwang, C. Y., Rhee, T. S., Snider, J. R., McMeeking, G. R., Dhaniyala, S., Lewis, E. R., Wentzell, J. J. B., Abbatt, J., Lee, C., Sultana, C. M., Ault, A. P., Axson, J. L., Diaz Martinez, M., Venero, I., Santos-Figueroa, G., Stokes, M. D., Deane, G. B., Mayol-Bracero, O. L., Grassian, V. H., Bertram, T. H., Bertram, A. K., Moffett, B. F. and Franc, G. D.: Sea spray aerosol as a unique source of ice nucleating particles, Proc. Natl. Acad. Sci., 113(21), 5797–5803, doi:10.1073/pnas.1514034112, 2016.

Emersic, C., Connolly, P. J., Boult, S., Campana, M. and Li, Z.: Investigating the discrepancy between wet-suspension- and dry-dispersion-derived ice nucleation efficiency of mineral particles, Atmos. Chem. Phys., 15(19), 11311–11326, doi:10.5194/acp-15-11311-2015, 2015.

Hader, J. D., Wright, T. P. and Petters, M. D.: Contribution of pollen to atmospheric ice nuclei concentrations, Atmos. Chem. Phys., 14(11), 5433–5449, doi:10.5194/acp-14-5433-2014, 2014.

[revised manuscript text omitted]
 the dry particles; $gf$ is the hygroscopic growth factor. The hygroscopic growth factor was based on the numerical model developed by Ming and Russell (2001) assuming the sampled aerosol consisted of sea spray aerosol with a 30 % organic mass content, following the assumption made in DeMott et al. (2016). This assumption results in growth factors consistent with measurements in the marine boundary layer (Zhou et al., 2001). For the density of the dry particles, we also assumed a sea spray aerosol with a 30 % organic mass content, resulting in a dry density of $1.87\ \mathrm{g\,cm^{-3}}$. To determine the sensitivity of the size distribution to the assumed composition of the aerosol, calculations were also carried out assuming a sea spray aerosol with a 10 % organic mass content and a 50 % organic mass content. The difference in the resulted size distributions assuming 10 %, 30 %, and 50 % organic mass content is small (see Fig. S7); hence, data shown in the main text only correspond to an assumed composition of a sea spray aerosol with a 30 % organic mass content.

**S2 Conversion of mobility diameter to aerodynamic diameter and correction for hygroscopic growth at Amphitrite Point**

At Amphitrite Point, dryers were used prior to sampling with the SMPS. As a result, SMPS data needs to be corrected for hygroscopic growth, and the mobility diameter needs to be converted to aerodynamic diameter. The equation to correct for hygroscopic growth is the following:

$$D_{m,RH} = gf \times D_{m,dry}, \qquad\qquad\qquad (S3)$$

where, $D_{m,RH}$ is the mobility diameter at the sampling RH; $D_{m,dry}$ is the mobility diameter under dry condition. The relationship between mobility diameter and aerodynamic diameter is given in Eq. (S1). Combining Eq. (S1) and Eq. (S3) results in the following:

$$D_{ae,RH} = gf \sqrt{\frac{\rho_{p,RH}}{\chi \rho_o}} D_{m,dry} = x D_{m,dry} \tag{S4}$$

where $D_{ae,RH}$ is the aerodynamic diameter at the sampling RH, and $x = gf \sqrt{\frac{\rho_{p,RH}}{\chi \rho_o}}$.

Equation (S4) illustrates that the relationship between the dry mobility diameter and the wet aerodynamic diameter is a simple factor $x$. To determine $x$, we varied this factor until the optimum fit was obtained between the SMPS and the APS data where overlap occurred (0.7 to 0.93 µm). This type of approach has been used successfully in the past to merge the SMPS and APS data (Beddows et al., 2010; Khlystov et al., 2004). Note, we did not use this approach in Sect. S1 since there was no overlapping size range between the SMPS and APS data measured at Labrador Sea and Lancaster Sound to allow an optimization of the fit.

**Table S1. The correction factors $f_{nu,1\,mm}$ and $f_{nu,0.25-0.1\,mm}$ for MOUDI stages 2-8 when using substrate holders. The uncertainty in $f_{nu,1\,mm}$ is given as the standard deviation.**

| MOUDI Stages | $f_{nu,1\,mm}$ | $f_{nu,0.25-0.1\,mm}$ |
|---|---|---|
| 2 | 0.74, +0.18, −0.12 | $0.1225\exp(-11.29\mu)+1.065\exp(-0.06412\mu)$ |
| 3 | 0.72, +0.08, −0.08 | $0.04718\exp(-14.15\mu)+1.023\exp(-0.02347\mu)$ |
| 4 | 1.18, +0.09, −0.14 | $0.04252\exp(-13.06\mu)+1.024\exp(-0.02386\mu)$ |
| 5 | 0.97, +0.03, −0.10 | $0.03023\exp(-14.97\mu)+1.015\exp(-0.01515\mu)$ |
| 6 | 0.75, +0.19, −0.02 | $0.5799\exp(-10.57\mu)+1.148\exp(-0.1408\mu)$ |
| 7 | 0.84, +0.07, −0.11 | $0.1151\exp(-10.66\mu)+1.072\exp(-0.07029\mu)$ |
| 8 | 1.01, +0.03, −0.12 | $1.03\exp(-12.79\mu)+1.268\exp(-0.2422\mu)$ |

$\mu = \frac{N_u(T)}{N_0}$, where $N_u(T)$ is the number of unfrozen droplets at temperature T, and $N_0$ is the total number of droplets in one freezing experiment.

[Figure]

[Figure]

altitude (m)

**Figure S1. The 3-day HYSPLIT back trajectories initiating at 50 m above ground level for Amphitrite Point (red dot), Labrador Sea (green dot) and Lancaster Sound (yellow dot). The back trajectories were calculated for every hour during the MOUDI sampling period. The starting points are labeled as coloured dots, and the altitude is shown using a colour scale.**

[Figure]

**Figure S2. The 3-day HYSPLIT back trajectories initiating at 150 m above ground level for Amphitrite Point (red dot), Labrador Sea (green dot) and Lancaster Sound (yellow dot). The back trajectories were calculated for every hour during the MOUDI sampling period. The starting points are labeled as coloured dots, and the altitude is shown using a colour scale.**

[Figure]

Figure S3. Comparison of aerosol particle number and surface area size distributions measured in this study (a, b) with aerosol particle number and surface area size distributions measured at a mid-latitude North-Atlantic marine boundary layer site (c, d) (O'Dowd et al., 2001). Case 1 from O'Dowd et al. (2001) corresponds to clean marine air measured under moderate humidity (80 %) and wind speeds (6 m s$^{-1}$) conditions, Case 2 corresponds to anthropogenically influenced maritime air measured at wind speeds in the order of 2-4 m s$^{-1}$, and Case 3 corresponds to anthropogenically influenced maritime air during a nucleation burst measured at wind speeds of 4 m s$^{-1}$.

[Figure]

**Figure S4.** Concentrations of (a) aerosol number, $N$, and (b) surface area, $S$, as a function of aerodynamic diameter, $D_{ae}$, using the same bin widths as the MOUDI. Each data point was calculated by averaging the numbers from Fig. 4 that were within the corresponding size bin. The x-error bars represent the widths of the size bins, and the y-error bars are propagated uncertainties from the error bars in Fig. 4. In most cases, the y-error bars are smaller than the size of the symbols.

| Deleted: 3 |
| Formatted: Font:Italic |
| Formatted: Font:Italic |
| Deleted: adding together |

[Figure]

Figure S5. $n_s$ values of sea spray aerosol as a function of temperature taken from DeMott et al. (2016). Shown is a linear fit to the data and 95% prediction bands.

[Figure]

Figure S6. $n_s$ values of mineral dust as a function of temperature taken from Niemand et al. (2012). Shown is a linear fit to the data and 95% prediction bands.

[Figure]

**Figure S7. Number size distribution measured at Labrador Sea (a) and Lancaster Sound (b) with the mobility diameter measured by the SMPS converted to aerodynamic diameter assuming a sea spray aerosol with 10 %, 30 %, and 50 % organic mass content.**

**S1  Corrections for hygroscopic growth**

A dryer was used prior to sampling with the SMPS at the Amphitrite Point site. To allow comparison with other measurements, the SMPS data at Amphitrite Point were corrected for hygroscopic growth using the following equation (Hämeri et al., 2000):

$$gf(RH) = \frac{D_{p,RH}}{D_{p,dry}},$$                                                                   (S1)

where *gf (RH)* is the hygroscopic growth factor at measured relative humidity (RH); $D_{p,RH}$ is the particle diameter at measured RH;  $D_{p,dry}$ is the dry particle diameter. The hygroscopic growth factor was calculated with the numerical model developed by Ming and Russell (2001) assuming the sampled aerosol consisted of sea spray aerosol with a 30% organic mass content, following the assumption made in DeMott et al. (2016). This assumption results in growth factors consistent with measurements in the marine boundary layer (Zhou et al., 2001) and a hygroscopicity parameter, κ, consistent with measurements at Amphitrite Point during the same campaign (Yakobi-Hancock et al., 2014).

---

## Author Response (AR2)

Prof. Paul Zieger
Co-Editor of Atmospheric Chemistry and Physics

Dear Paul,

Listed below are our responses to the comments from the referees of our manuscript. For clarity and visual distinction, the referee comments or questions are listed here in black and are preceded by bracketed, italicized numbers (e.g. *[1]*). Author's responses are offset in blue below each referee statement with matching numbers (e.g. *[A1]*). We thank the referees for carefully reading our manuscript and for their helpful comments!

Sincerely,

Allan Bertram,
Professor, Department of Chemistry
University of British Columbia

**Referee #1:**

"Ice-nucleating ability of aerosol particles and possible sources at three coastal marine sites" by Meng Si et al, resubmitted to ACPD

The manuscript has been improved to such an extent that it is now close to merit publication in ACP. However, I still have a few remarks I'll give below. But I trust that the authors will consider them carefully and will therefore only suggest minor revisions (although I feel that these remarks are still important). Much of this deals with how you treat / mix different concepts of particle size and related surface area (dry / wet; geometric / aerodynamic; mineral dust fraction / total aerosol). I suggest a few estimations that can at least put these different "approaches" into perspective with each other. They should be included in the text.

*[1]* 1.) Typically when measuring particle number size distributions in the atmosphere, the dry particle size is used as a reference (and often also the geometric diameter instead of the aerodynamic one), and typically this is also what should be used for the derivation of other values as e.g., surface area distributions, $n_s$ and such. This is done to be comparable, as otherwise reported results change with the RH at which the measurement was done and cannot be easily compared. I'm sorry that I did not pick that up so clearly in the first round, as I feel this could have been easily corrected. (And you stumbled across that yourself when you compare with $n_s$ derived from DeMott et al. (2016), top line of page 11.)

For reasons of comparability, the authors could consider changing this, although it is clear to me that this is requesting quite a bit as it needs to do a lot of recalculation. This is also why the authors could also settle for the second best solution which would be to deal with this openly. This means, that it should be estimated by how much the "wet $n_s$

related to aerodyn. diameter" you derive underestimates the "n_s based on dry particle diameters" (it would be good to also convert aerodynamic to geometric diameter, for that exercise, too). This is easily done, following the conversion you describe in the SI and deriving the surface area for the different bins under these circumstances (the MOUDI-stage-cut-offs would have to be converted to geometric diameters, too, then, if you used geometric diameters to calculate the surface area).

This could quite quickly give you a measure of how much you might underestimate n_s, and that could be added to the text (in sec. 2.4, for example, where you discuss the conversion, or wherever you feel it fits). This can then also be used in the argument when the comparison with the marine INP concentrations from DeMott et al. (2016) is done, although you quite correctly say that the DeMott-based n_s values are an upper limit. (BTW: are n_s values from DeMott et al. (2016) related to the surface area of the total aerosol or only he sea spray particles? Check that, and if the latter is the case, the same applies as for the comparison with mineral dust that I comment on in the remark 2.)

*[A1]* We agree with the referee that it would be better to present the results using the dry-geometric diameter rather than the wet-aerodynamic diameter, if possible. To convert from wet-aerodynamic diameter to dry-geometric diameter, information on the chemical composition of the particles is needed. We can make reasonable assumptions about the chemical composition of the total aerosol, allowing us to convert the wet-aerodynamic diameter of the total aerosol to the dry-geometric diameter of the total aerosol, or *vice versa*. Unfortunately, we do not feel justified making assumptions about the chemical composition of the INPs *a priori* (in fact, one of the goals of the manuscript is to provide insight on the chemical composition of the INPs). As a result, we do not feel justified converting our INP results from wet-aerodynamic diameter to dry-geometric diameter. Luckily, the main conclusions in our manuscript can be made using either dry-geometric diameter or wet-aerodynamic diameter.

Nevertheless, to address the referee's comments as best as we can, we have taken the second best solution the referee suggested, and estimated how much the $n_s$ values based on dry-geometric diameter deviated from $n_s$ values based on wet-aerodynamic diameter for both sea spray aerosol and mineral dust. This information has been added to Section 3.5 where we compared our results with $n_s$ values from the literature. The details have been added as Section S3 in the Supplement. Specifically, the following sentences have been added to the revised manuscript:

"…Since the reported $n_s$ values in DeMott et al. (2016) were based on dry, geometric diameters, they overestimate the $n_s$ values based on wet, aerodynamic diameters at 95 % RH by a factor of 6 (see Sect. S3 in the Supplement). … Since the reported $n_s$ values in Niemand et al. (2012) were based on geometric diameters, they overestimate the $n_s$ values based on aerodynamic diameters by a factor of 2 (see Sect. S3 in the Supplement). …"

*[2]* And a related point is that the hygroscopic growth factors should be explicitly mentioned in the SI (only saying "factors consistent with measurements in the marine boundary layer" makes them less trustworthy than they likely are).

*[A2]* The hygroscopic growth factors have been added explicitly to Section S1 in the revised Supplement.

*[3]* 2.) On page 10, lines 18-19, you say:" Since this equation considers the surface area of all aerosol particles, rather than the surface area of just the INPs, the calculated ns values are lower limits to the ns values for the INPs."

Comparing n_s derived based on mineral dust particles only with n_s derived from the total surface area of the measured aerosol is a bit like comparing apples and oranges. And using the total surface area of the measured aerosol is a good way to go. (This ultimately is what the model output gives you, too, or am I mistaken? As does the parameterization suggested by DeMott et al., 2010.) It just needs to be well defined, and then comparisons of values based on different bases should be avoided.

In that sense, it does not make sense if you talk about a "lower limit". "Lower limit" connected to what? Make clear that you compare different parameters, where one is a value retrieved from laboratory measurement relating to mineral dust INP only, while the other one relates to the total atmospheric aerosol.

*[A3]* To address the referee's comment we have modified the sentence mentioned by the referee to the following:

"Since this equation considers the surface area of all aerosol particles, rather than the surface area of just the INPs, the calculated $n_s$ values correspond to the total atmospheric aerosol."

*[4]* It would be interesting if you could give an order of magnitude by which n_s based on only mineral dust and "all aerosols" deviate. I guess you could at least estimate this number roughly, based on your data. Simply calculate the surface area related to the INP concentrations and compare that to the surface area you show in Fig. 4 (b) - the resulting factor is the one by which n_s related to "all aerosols" differs from that related to "only INP" - and that at least should give an idea about the order of magnitude of the deviation, and could help putting your n_s in relation to that derived from Niemand et al. (2012).

DeMott, P. J., A. J. Prenni, X. Liu, S. M. Kreidenweis, M. D. Petters, C. H. Twohy, M. S. Richardson, T. Eidhammer, and D. C. Rogers (2010), Predicting global atmospheric ice nuclei distributions and their impact on climate, Proc. Natl. Acad. Sci. USA, 107(25), 11217-11222, doi:10.1073/pnas.0910818107.

*[A4]* To address the referee's comment, in the revised manuscript we have estimated how much $n_s$ based on only mineral dust will change assuming that mineral dust accounts for 50% of the total aerosol surface area measured. This estimation gives a rough idea of how $n_s$ based on only mineral dust and $n_s$ based on the total atmospheric aerosol deviate. The following is the specific text that has been added to the revised manuscript:

"If we assume mineral dust particles are the only INPs in the atmosphere, and they account for 50 % of the total aerosol surface area, then the $n_s$ values of mineral dust shown in Fig. 7 divided by a factor of 2 would correspond to the $n_s$ values of the total atmospheric aerosol."

3) A few specific remarks:

*[5]* page 11, line 6-7: "sea spray aerosol was … not a contributor to the largest INPs (5.6-10 μm in size) observed at Lancaster Sound." - You can really only say that it did not contribute the majority of these INP! There could still be some sea spray particles that act as INP in this size range. Please adjust the sentence.

*[A5]* In the revised manuscript, the sentence has been adjusted to "sea spray aerosol was… not a major contributor to the largest INPs (5.6-10 μm in size) observed at Lancaster Sound".

*[6]* page 11, line 18: I assume when you talk about "measured INP concentration" here, this is the sum of the concentrations in all size ranges? Mention this!

*[A6]* In the revised manuscript, the sentence has been changed to "…comparison between the measured total INP concentrations (sum of the INP concentrations for all sizes measured) and the simulated INP concentrations at the surface at the three sites…"

*[7]* page 12, end of section 3.5: Deviations between measurements and model results can also come from uncertainties in the measurements. You may have losses in the MOUDI (the RH of 50% that you give is comparably low alreaedy, and you'll get the 50% only, at the described circumstances, as a possible maximum value, anyway (dew point outside 14°C, T in the container 25°C)), and you may overestimate the surface area and underestimate n_s (as you use the wet aerodynamic diameter – my assumption would be that the model used dry geometric diameters). This should be shortly discussed in a new paragraph.

*[A7]* In the revised manuscript, to address the referee's comment, we have added the following to Section 3.6:

"As discussed in Section 2.1.1, particle rebound from the substrate can be an issue when sampling particles with an inertia impactor. Good agreement between INP concentrations measured by the MOUDI-DFT and INP concentrations measured by a continuous flow diffusion chamber (a technique that is not susceptible to rebound) has been observed in previous field campaigns when the RH of the sampled aerosol stream was as low as 40-45 % (DeMott et al., 2017; Mason et al., 2015b). Nevertheless, particle rebound cannot be completely ruled out in the current study. If particle rebound was a factor when collecting particles with the MOUDI in the current study, the measured INP concentrations would be lower limits to the true INP concentrations, and the differences between simulated INP concentrations and measured INP concentrations shown in Fig. 8 would only be larger."

*[8]* page 12, line 20 ff: This example given here is very arbitrary. Particularly for black carbon, there was a number of indications lately that these do not act as INP, e.g., https://meetingorganizer.copernicus.org/EGU2017/EGU2017-16057.pdf, and (already published) Chen et al. (2018).

Also, this would better be moved to section 3.5 and discussed there, where it is first speculated on the four different types (as I wondered there what you might imagine them to be, anyway). As it was not mentioned before, it is not suited to appear at the "Summary and Conclusion". And if you leave the choice of the four types, this information that BC might not be a good / no INP should be added, too.

Chen, J., Z. Wu, S. Augustin-Bauditz, S. Grawe, M. Hartmann, X. Pei, Z. Liu, D. Ji, and H. Wex (2018), Ice nucleating particle concentrations unaffected by urban air pollution in Beijing, China, Atmos. Chem. Phys., 18, 3523–3539, doi:10.5194/acp-18-3523-2018.

*[A8]* To address the referee's comment, the example of the four different types has been moved from "Summary and Conclusions" to Section 3.5. Also, the following sentence has been added to Section 3.5:

"The assumption of a small $n_s$ for black carbon internally mixed with sulfate aerosols is consistent with previous measurements (e.g., Brooks et al., 2014; Chen et al., 2018)."

**Referee #2:**

Si et al. have suitably revised their manuscript discussing size-resolved INP properties at three coastal sites through observations and model-observational comparisons. I have a few minor comments, after which the manuscript should be accepted for publication in ACP.

*[9]* P2, l14: Also, wave breaking.

*[A9]* In the revised manuscript, the sentence has been changed to "these INPs in seawater are thought to be emitted into the atmosphere by wave breaking and bubble bursting mechanisms…"

*[10]* Section 2.3.2: So, was the Amundsen stationary during the days the samples were collected or was in it transit? This is not clear. If it was in transit, a ship track or some sort of information on the spatial extent of the sampling time period should be provided.

*[A10]* The Amundsen was in transit while the samples were collected. The change of the coordinates was less than 0.5 degree in longitude, and less than 1 degree in latitude during the sampling period. The following sentence has been added to Section 2.3.2 in the revised manuscript:

"During both sample collection periods, the Amundsen was in transit, and the change of the coordinates was less than 0.5 degree in longitude, and less than 1 degree in latitude."

*[11]* P8, l13-15: The sentences describing the initiation heights are redundant from the last section. The authors could simply state that 50 and 150 m.a.g.l. are shown in the supporting information.

*[A11]* In the revised manuscript, the sentence describing the initiation heights in Section 3.1 has been deleted to avoid redundancy.

*[12]* P9, l27-28: It is not clear how one diameter value is indicative of the size bins for the APS, SMPS, and especially INPs. Is this midpoint or starting diameter for each bin? Either way, how was the INP single diameter value determined?

*[A12]* The midpoint of each size bin was used as the size of aerosol particles and INPs measured by the APS, SMPS and MOUDI. This has been made clear in the figure captions in the revised manuscript.

*[13]* Table 1: Either ranges (min – max) or standard deviation should be provided for the averages such that the variability of sampling time and meteorological parameters is evident.

*[A13]* The standard deviation has been added to the average as the variability of the parameters shown in Table1.

[revised manuscript text omitted]

$$\rho_{p,RH} = \rho_w + (\rho_{p,dry} - \rho_w)\frac{1}{gf^3}, \tag{S2}$$

where $\rho_w$ is the density of water; $\rho_{p,dry}$ is the density of the dry particles; $gf$ is the hygroscopic growth factor. The hygroscopic growth factor was based on the numerical model developed by Ming and Russell (2001) assuming the sampled aerosol consisted of sea spray aerosol with a 30 % organic mass content, following the assumption made in DeMott et al. (2016). This assumption results in growth factors of 1.2 at 70 % RH, and 2.4 at 95 % RH, which are consistent with measurements made in the Arctic summer marine boundary layer by Zhou et al. (2001) (1.23 ± 0.09 at 70 % RH, 2.05 ± 0.11 at 90 % RH). For the density of the dry particles, we also assumed a sea spray aerosol with a 30 % organic mass content, resulting in a dry density of 1.87 g cm$^{-3}$. To determine the sensitivity of the size distribution to the assumed composition of the aerosol, calculations were also carried out assuming a sea spray aerosol with a 10 % organic mass content and a 50 % organic mass content. The difference in the resulted size distributions assuming 10 %, 30 %, and 50 % organic mass content is small (see Fig. S7); hence, data shown in the main text only correspond to an assumed composition of a sea spray aerosol with a 30 % organic mass content.

**S2 Conversion of mobility diameter to aerodynamic diameter and correction for hygroscopic growth at Amphitrite Point**

At Amphitrite Point, dryers were used prior to sampling with the SMPS. As a result, SMPS data needs to be corrected for hygroscopic growth, and the mobility diameter needs to be converted to aerodynamic diameter. The equation to correct for hygroscopic growth is the following:

$$D_{m,RH} = gf \times D_{m,dry}, \tag{S3}$$

Meng Si 2018-8-3 3:56 PM

Meng Si 2018-8-3 3:59 PM

where, $D_{m,RH}$ is the mobility diameter at the sampling RH; $D_{m,dry}$ is the mobility diameter under dry condition. The relationship between mobility diameter and aerodynamic diameter is given in Eq. (S1). Combining Eq. (S1) and Eq. (S3) results in the following:

$$D_{ae,RH} = gf \sqrt{\frac{\rho_{p,RH}}{\chi \rho_o}} D_{m,dry} = x D_{m,dry},$$ (S4)

5 where $D_{ae,RH}$ is the aerodynamic diameter at the sampling RH, and $x = gf \sqrt{\frac{\rho_{p,RH}}{\chi \rho_o}}$.

Equation (S4) illustrates that the relationship between the dry mobility diameter and the wet aerodynamic diameter is a simple factor $x$. To determine $x$, we varied this factor until the optimum fit was obtained between the SMPS and the APS data where overlap occurred (0.7 to 0.93 μm). This type of approach has been used successfully in the past to merge the SMPS and APS data (Beddows et al., 2010; Khlystov et al., 2004). Note, we did not use this approach in Sect. S1 since there

10 was no overlapping size range between the SMPS and APS data measured at Labrador Sea and Lancaster Sound to allow an optimization of the fit.

**S2 Conversion of $n_s$ values based on dry, geometric diameters to $n_s$ values based on wet, aerodynamic diameters**

The $n_s$ values of sea spray aerosol reported in DeMott et al. (2016) and the $n_s$ values of mineral dust reported in Niemand et al. (2012) are based on dry, geometric diameters. In the following, we investigated how much $n_s$ values based on dry,

15 geometric diameters ($n_{s\_geo,dry}$) overestimate $n_s$ values based on wet, aerodynamic diameters ($n_{s\_ae,RH}$).

Assuming the particles are all spherical, the mobility diameter of a particle is the same as its geometric diameter. Thus, Eq. (S4) can be written as the following:

$$D_{ae,RH} = gf \sqrt{\frac{\rho_{p,RH}}{\chi \rho_o}} D_{geo,dry} = x D_{geo,dry},$$ (S5)

where $D_{geo,dry}$ is the dry, geometric diameter, and $x = gf \sqrt{\frac{\rho_{p,RH}}{\chi \rho_o}}$.

20 The $n_s$ values based on wet, aerodynamic diameters can be calculated using the following equation:

$$n_{s\_ae,RH} = \frac{[INPs]}{S_{tot,ae,RH}} = \frac{[INPs]}{\pi D_{ae,RH}^2 N_{tot}} = \frac{[INPs]}{\pi x^2 D_{geo,dry}^2 N_{tot}} = \frac{[INPs]}{x^2 S_{tot,geo,dry}} = \frac{n_{s\_geo,dry}}{x^2},$$ (S6)

where $[INPs]$ is the concentration of INPs; $S_{tot,ae,RH}$ is the total surface area based on wet, aerodynamic diameters; $N_{tot}$ is the total number of aerosol particles, $S_{tot,geo,dry}$ is the total surface area based on dry, geometric diameters.

According to Eq. (6), $n_s$ values based on dry, geometric diameters overestimate $n_s$ values based on wet, aerodynamic by a

25 factor of $x^2$.

For sea spray aerosol, we assumed a 30 % organic mass content, which resulted in $gf$ of 2.4, and $\rho_{p,RH}$ of 1.1 g cm$^{-3}$ at the highest sampling RH (95 %) in this study. For these conditions, $x^2$ is approximately 6.

Meng Si 2018-8-8 6:13 PM

Meng Si 2018-8-8 5:55 PM

Meng Si 2018-8-8 6:13 PM

Meng Si 2018-8-8 6:14 PM

Meng Si 2018-8-8 6:14 PM

For mineral dust, we assumed mineral dust is non-hygroscopic (therefore $gf$ is 1) with a density of 2 g cm$^{-3}$ (Khlystov et al., 2004). For these conditions, $x^2$ is 2.

**References:**

[revised manuscript text omitted]

---

## Author Response (AR3)

Prof. Paul Zieger
Co-Editor of Atmospheric Chemistry and Physics

Dear Paul,

Listed below are our responses to your comments. For clarity and visual distinction, the comments are listed here in black and are preceded by bracketed, italicized numbers (e.g. *[1]*). Author's responses are in blue below each comment with matching numbers (e.g. *[A1]*). Thank you for carefully reading our manuscript and for the helpful comments!

Sincerely,

Allan Bertram,
Professor, Department of Chemistry
University of British Columbia

*[1]* Sect 2.2. Please state if the SMPS instruments were operated with an impactor to remove doubly and triply charged particles (which can become important when converting to surface size distributions).

*[A1]* The SMPS was operated with a built in inertial impactor at the inlet to remove large particles outside the measurement range that may contribute to errors caused by multiple charging. This information has been added to Section 2.2 in the manuscript.

*[2]* Figure 5: Please improve the y-label and add a proper unit label. I guess you mean divide by a concentration (INP/N_total)?

*[A2]* Thank you for the suggestion. The y-label has been changed to $INP/N_{aerosol}$, where $N_{aerosol}$ represents the number of aerosol particles in a given size bin. The figure caption has been adjusted accordingly.

*[3]* Figure 4, S3 and S7: I may have missed it: Is this a cumulative size distribution or an average one for the respective observation periods? Please clarify (in text and caption).

*[A3]* The size distribution is an average size distribution for the sampling period. This has been clarified in Section 3.3 and in the figure captions.

*[4]* Table S1: Please explain in the caption, on why there are three values for the first correction factor.

*[A4]* The three values are the mean value and the uncertainties. The caption has been adjusted to make this clear.

[revised manuscript text omitted]
 the dry particles; $gf$ is the hygroscopic growth factor. The hygroscopic growth factor was based on the numerical model developed by Ming and Russell (2001) assuming the sampled aerosol consisted of sea spray aerosol with a 30 % organic mass content, following the assumption made in DeMott et al. (2016). This assumption results in growth factors of 1.2 at 70 % RH, and 2.4 at 95 % RH, which are consistent with measurements made in the Arctic summer marine boundary layer by Zhou et al. (2001) ($1.23 \pm 0.09$ at 70 % RH, $2.05 \pm 0.11$ at 90 % RH). For the density of the dry particles, we also assumed a sea spray aerosol with a 30 % organic mass content, resulting in a dry density of $1.87\ \mathrm{g\,cm^{-3}}$. To determine the sensitivity of the size distribution to the assumed composition of the aerosol, calculations were also carried out assuming a sea spray aerosol with a 10 % organic mass content and a 50 % organic mass content. The difference in the resulted size distributions assuming 10 %, 30 %, and 50 % organic mass content is small (see Fig. S7); hence, data shown in the main text only correspond to an assumed composition of a sea spray aerosol with a 30 % organic mass content.

**S2 Conversion of mobility diameter to aerodynamic diameter and correction for hygroscopic growth at Amphitrite Point**

At Amphitrite Point, dryers were used prior to sampling with the SMPS. As a result, SMPS data needs to be corrected for hygroscopic growth, and the mobility diameter needs to be converted to aerodynamic diameter. The equation to correct for hygroscopic growth is the following:

$$D_{m,RH} = gf \times D_{m,dry}, \tag{S3}$$

where, $D_{m,RH}$ is the mobility diameter at the sampling RH; $D_{m,dry}$ is the mobility diameter under dry condition. The relationship between mobility diameter and aerodynamic diameter is given in Eq. (S1). Combining Eq. (S1) and Eq. (S3) results in the following:

$$D_{ae,RH} = gf\sqrt{\frac{\rho_{p,RH}}{\chi\rho_o}}D_{m,dry} = xD_{m,dry},$$ (S4)

5    where $D_{ae,RH}$ is the aerodynamic diameter at the sampling RH, and $x = gf\sqrt{\frac{\rho_{p,RH}}{\chi\rho_o}}$.

Equation (S4) illustrates that the relationship between the dry mobility diameter and the wet aerodynamic diameter is a simple factor $x$. To determine $x$, we varied this factor until the optimum fit was obtained between the SMPS and the APS data where overlap occurred (0.7 to 0.93 µm). This type of approach has been used successfully in the past to merge the SMPS and APS data (Beddows et al., 2010; Khlystov et al., 2004). Note, we did not use this approach in Sect. S1 since there

10   was no overlapping size range between the SMPS and APS data measured at Labrador Sea and Lancaster Sound to allow an optimization of the fit.

**S3 Conversion of $n_s$ values based on dry, geometric diameters to $n_s$ values based on wet, aerodynamic diameters**

The $n_s$ values of sea spray aerosol reported in DeMott et al. (2016) and the $n_s$ values of mineral dust reported in Niemand et al. (2012) are based on dry, geometric diameters. In the following, we investigated how much $n_s$ values based on dry,

15   geometric diameters ($n_{s\_geo,dry}$) overestimate $n_s$ values based on wet, aerodynamic diameters ($n_{s\_ae,RH}$).

Assuming the particles are all spherical, the mobility diameter of a particle is the same as its geometric diameter. Thus, Eq. (S4) can be written as the following:

$$D_{ae,RH} = gf\sqrt{\frac{\rho_{p,RH}}{\chi\rho_o}}D_{geo,dry} = xD_{geo,dry},$$ (S5)

where $D_{geo,dry}$ is the dry, geometric diameter, and $x = gf\sqrt{\frac{\rho_{p,RH}}{\chi\rho_o}}$.

20   The $n_s$ values based on wet, aerodynamic diameters can be calculated using the following equation:

$$n_{s\_ae,RH} = \frac{[INPs]}{S_{tot,ae,RH}} = \frac{[INPs]}{\pi D_{ae,RH}^2 N_{tot}} = \frac{[INPs]}{\pi x^2 D_{geo,dry}^2 N_{tot}} = \frac{[INPs]}{x^2 S_{tot,geo,dry}} = \frac{n_{s\_geo,dry}}{x^2},$$ (S6)

where $[INPs]$ is the concentration of INPs; $S_{tot,ae,RH}$ is the total surface area based on wet, aerodynamic diameters; $N_{tot}$ is the total number of aerosol particles, $S_{tot,geo,dry}$ is the total surface area based on dry, geometric diameters.

According to Eq. (S6), $n_s$ values based on dry, geometric diameters overestimate $n_s$ values based on wet, aerodynamic by a

25   factor of $x^2$.

For sea spray aerosol, we assumed a 30 % organic mass content, which resulted in $gf$ of 2.4, and $\rho_{p,RH}$ of 1.1 g cm$^{-3}$ at the highest sampling RH (95 %) in this study. For these conditions, $x^2$ is approximately 6.

For mineral dust, we assumed mineral dust is non-hygroscopic (therefore $gf$ is 1) with a density of 2 g cm$^{-3}$ (Khlystov et al., 2004). For these conditions, $x^2$ is 2.

**References:**

[revised manuscript text omitted]

Meng Si 2018-10-4 2:46 PM